# Rapid Spatio-Temporal Flood Modelling via Hydraulics-Based Graph Neural Networks

Roberto Bentivoglio[1], Elvin Isufi[2], Sebastiaan Nicolas Jonkman[3], and Riccardo Taormina[1]

[1]Department of Water Management, Faculty of Civil Engineering and Geosciences, Delft University of Technology, Delft, The Netherlands
[2]Department of Intelligent Systems, Faculty of Electrical Engineering, Mathematics and Computer Science, Delft University of Technology, Delft, The Netherlands
[3]Department of Hydraulic Engineering, Faculty of Civil Engineering and Geosciences, Delft University of Technology, Delft, The Netherlands

**Correspondence:** Roberto Bentivoglio, r.bentivoglio@tudelft.nl

**Abstract.** Numerical modelling is a reliable tool for flood simulations, but accurate solutions are computationally expensive. In the recent years, researchers have explored data-driven methodologies based on neural networks to overcome this limitation. However, most models are used only for a specific case study and disregard the dynamic evolution of the flood wave. This limits their generalizability to topographies that the model was not trained on and in time-dependent applications. In this paper, we introduce SWE-GNN, a hydraulics-inspired surrogate model based on Graph Neural Networks (GNN) that can be used for rapid spatio-temporal flood modelling. The model exploits the analogy between finite volume methods, used to solve the shallow water equations (SWE), and GNNs. For a computational mesh, we create a graph by considering finite-volume cells as nodes and adjacent cells as connected by edges. The inputs are determined by the topographical properties of the domain and the initial hydraulic conditions. The GNN then determines how fluxes are exchanged between cells via a learned local function. We overcome the time-step constraints by stacking multiple GNN layers, which expand the considered space instead of increasing the time resolution. We also propose a multi-step-ahead loss function along with a curriculum learning strategy to improve the stability and performance. We validate this approach using a dataset of two-dimensional dike breach flood simulations on randomly-generated digital elevation models, generated with a high-fidelity numerical solver. The SWE-GNN model predicts the spatio-temporal evolution of the flood for unseen topographies with a mean average error in time of $0.04\ m$ for water depths and $0.004\ m^2/s$ for unit discharges. Moreover, it generalizes well to unseen breach locations, bigger domains, and over longer periods of time, compared to those of the training set, outperforming other deep learning models. On top of this, SWE-GNN has a computational speedup of up to two orders of magnitude faster than the numerical solver. Our framework opens the doors to a new approach for replacing numerical solvers in time-sensitive applications with spatially-dependant uncertainties.

# 1 Introduction

Accurate flood models are essential for risk assessment, early warning, and preparedness for flood events. Numerical models can characterize how floods evolve in space and time, with the two-dimensional (2D) hydrodynamic models being the most popular (Teng et al., 2017). They solve a discretized form of the depth-averaged Navier-Stokes equations, referred to as Shallow Water Equations (SWE) (Vreugdenhil, 1994). Numerical models are computationally expensive, making them inapplicable for real-time emergencies and uncertainty analyses. Several methods aim to speed up the solution of these equations by either approximating them (Bates and De Roo, 2000) or by using high-performance computing and parallelization techniques (Hu et al., 2022; Petaccia et al., 2016). However, approximate solutions are valid only for domains with low spatial and temporal gradients (Costabile et al., 2017), while high-performance computing methods are bound by the numerical constraints and the computational resources.

Data-driven alternatives speed up numerical solvers (Mosavi et al., 2018). In particular, deep learning outperforms other machine learning methods used for flood modelling, in both speed and accuracy (Bentivoglio et al., 2022). Berkhahn et al. (2019) developed a multi-layer perceptron model for predicting urban floods given a rainfall event, achieving promising speed-ups and accuracy. Guo et al. (2021) and Kabir et al. (2020) developed convolutional neural networks (CNN) for river flood inundation, while Jacquier et al. (2021) used deep learning to facilitate the reduced order modelling of dam break floods and provide uncertainty estimates. Also Zhou et al. (2022) employed a CNN-based model to determine the spatio-temporal variation of flood inundation from a set of representative locations. These works explored the generalization of boundary conditions on a fixed domain, i.e., they changed the return period of the floods for a single case study, but they need retraining when applied to a new area, requiring more resources in terms of data, model preparation, and computation times.

To overcome this issue, the community is investigating the generalizability of deep learning models to different study areas. Löwe et al. (2021) proposed a CNN model to estimate the maximum water depth of pluvial urban floods. They trained their model on part of their case study and then deployed it on the unseen parts, showing consistent performances. Guo et al. (2022) accurately predicted the maximum water depth and flow velocities for river floods in different catchments in Switzerland. To incorporate the variations in catchment size and shape, they divided the domain into patches. do Lago et al. (2023) proposed a conditional generative adversarial networks that could predict the maximum water depth unseen rain events in unseen urban catchments. However, these approaches focus on a single maximum depth or velocity map, disregarding the dynamical behaviour, i.e., no information is provided on the flood conditions over space and time, which is crucial for evacuation and response to the flood.

To overcome this limitation, we propose SWE-GNN, a deep learning model merging graph neural networks (GNN) with the finite-volume methods used to solve the SWE. GNNs generalize convolutional neural networks to irregular domains such as graphs and have shown promising results for fluid dynamics (e.g., Lino et al., 2021; Peng et al., 2022) and partial differential equations (e.g., Brandstetter et al., 2022; Horie and Mitsume, 2022). Hence, developing GNNs that follow the SWE equations is not only more physically interpretable but also allows better generalization abilities to unseen flood evolution, unseen breach location and unseen topographies. In particular, we exploit the geometrical structure of the finite-volume computa-

tional mesh by using its dual graph, obtained by connecting the centres of neighbouring cells via edges. The nodes represent finite-volume cells and edges fluxes across them. Following an explicit numerical discretization of the SWE, we formulate a novel GNN propagation rule that learns how fluxes are exchanged between cells, based on the gradient of the hydraulic variables. We set the number of GNN layers based on the time step between consecutive predictions, in agreement with the Courant–Friedrichs–Lewy conditions. The inputs of the model are the hydraulic variables at a given time, elevation, slopes, area, length, and orientation of the mesh's cells. The outputs are the hydraulic variables at the following time step, evaluated in an auto-regressive manner, i.e., the model is repeatedly applied using its predictions as inputs to produce extended simulations.

We tested our model on dike-breach flood simulations due to their time-sensitive nature and presence of uncertainties in topography and breach formation (Jonkman et al., 2008; Vorogushyn et al., 2009). Moreover, given the sensibility to floods of low-lying areas, fast surrogate models that generalize over all those uncertainties are required for probabilistic analyses. By doing so, our key contributions are threefold:

– We develop a new graph neural network model where the propagation rule and the inputs are taken from the shallow water equations. In particular, the hydraulic variables propagate based on their gradient across neighbouring finite volume cells;

– We improve the model's stability by training it via a multi-step-ahead loss function, that results in stable predictions up to 120 hours ahead, using only the information of the first hour as initial hydraulic input;

– We show that the proposed model can surrogate numerical solvers for spatio-temporal flood modelling in unseen topographies and unseen breach locations, with two orders of magnitude speed-ups.

The rest of the paper is structured as follows: Section 2 illustrates the theoretical background; Section 3 describes the proposed methodology. In Section 4, we present the dataset used for the numerical experiments. Section 5 shows the results obtained with the proposed model and compares it with other deep learning models. Finally, Section 6 discusses the results, analyses the current limitations of this approach, and proposes future research directions.

## 2 Theoretical background

In this section, we describe the theory supporting our proposed model. First, we discuss numerical models for flood modelling; then, we present deep learning models, focusing on graph neural networks. Throughout the paper, we use the standard vector notation, with $a$ scalar, $\mathbf{a}$ vector, $\mathbf{A}$ matrix, and $\mathcal{A}$ tensor.

### 2.1 Numerical modelling

#### 2.1.1 Shallow water equations

When assuming negligible vertical accelerations, floods can be modelled via the shallow waters equations (SWE) (Vreugdenhil, 1994). These are a system of hyperbolic partial differential equations that describe the behaviour of shallow flows by enforcing

mass and momentum conservation. The two-dimensional SWE can be written as

$$\frac{\partial \mathbf{u}}{\partial t} + \nabla \mathbf{F} = \mathbf{s}, \tag{1}$$

with

$$\mathbf{u} = \begin{pmatrix} h \\ q_x \\ q_y \end{pmatrix}, \mathbf{F} = \begin{pmatrix} q_x & q_y \\ \frac{q_x^2}{h} + \frac{gh^2}{2} & \frac{q_x q_y}{h} \\ \frac{q_x q_y}{h} & \frac{q_y^2}{h} + \frac{gh^2}{2} \end{pmatrix}, \mathbf{s} = \begin{pmatrix} 0 \\ gh(s_{0x} - s_{fx}) \\ gh(s_{0y} - s_{fy}) \end{pmatrix}, \tag{2}$$

where $\mathbf{u}$ represents the conserved variable vector, $\mathbf{F}$ the fluxes in the $x$ and $y$ directions, and $\mathbf{s}$ the source terms. Here, $h[m]$ represents the water depth, $q_x = uh[m^2/s]$ and $q_y = vh[m^2/s]$ are the averaged components of the discharge vector along the $x$ and $y$ coordinates, respectively, and $g[m/s^2]$ is the acceleration of gravity. The source terms in $\mathbf{s}$ depend on the contributions of bed slopes $\mathbf{s}_0$ and friction losses $\mathbf{s}_f$ along the two coordinate directions.

### 2.1.2 Finite volume method

The SWE cannot be solved analytically unless some simplifications are enforced. Thus, they are commonly solved via spatio-temporal numerical discretizations, such as the finite volume method (e.g., Alcrudo and Garcia-Navarro, 1993). This method discretizes the spatial domain using meshes, i.e., geometrical structures composed of nodes, edges, and faces. We consider each finite volume cell is represented by its centre of mass, where the hydraulic variables, $h$, $q_x$, and $q_y$, are defined (see Fig. 1). The governing equations are then integrated over the cells, considering piece-wise constant variations, i.e., the value of the variables at a certain time instant is spatially uniform for every cell. The SWE can be discretized in several ways both in space and time (e.g., Petaccia et al., 2013; Xia et al., 2017) but we focus on a first-order explicit scheme with a generic spatial discretization. For an arbitrary volume $\Omega_i$ and a discrete time step $\Delta t$, the SWE (eq. (1)) can be re-written as:

$$\mathbf{u}_i^{t+1} = \mathbf{u}_i^t + \left( \mathbf{s}_i - \sum_{j=1}^{N_i} (\mathbf{F} \cdot \mathbf{n})_{ij} \frac{l_{ij}}{a_i} \right) \Delta t \tag{3}$$

with $\mathbf{u}_i^t$ the hydraulic variables at time $t$ and cell $i$, $a_i$ the area of the $i^{th}$ cell, $N_i$ the number of neighbouring cells, $l_{ij}$ the length of the $j^{th}$ side of cell $i$, $\mathbf{s}_i$ the source terms, $\mathbf{n}_{ij} = [n_{xij}, n_{yij}]$ the outward unit normal vector in the $x$ and $y$ directions for side $ij$, and $(\mathbf{F} \cdot \mathbf{n})_{ij}$ the numerical fluxes across neighbouring cells.

In numerical models with explicit discretization, stability is enforced by satisfying the Courant–Friedrichs–Lewy (CFL) condition, which imposes the numerical propagation speed to be lower than the physical one (Courant et al., 1967). Considering $v$ as propagation speed, the Courant number $C$ can be evaluated as

$$C = \frac{v \Delta t}{\Delta x}, \tag{4}$$

where $\Delta t$ and $\Delta x$ represent the time step and the mesh size. This condition forces $\Delta t$ to be sufficiently small, to avoid a too-fast propagation of water in space that would result in a loss of physical consistency. Small time steps imply increasing number

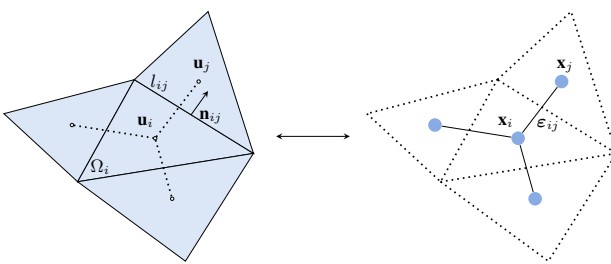

**Figure 1.** Schematic representation of an arbitrary triangular volume mesh and its dual graph. Left: a finite-volume cell $\Omega_i$ along with its neighboring cells. Vectors $\mathbf{u}_i$ and $\mathbf{u}_j$ represent the cells' hydraulic variables, while $l_{ij}$ and $\mathbf{n}_{ij}$ corresponds, respectively, to the length of the mesh side and the outward unit normal vector, between cells $i$ and $j$. Right: the dual graph of the mesh is obtained by considering each $i^{th}$ cell's center as a node $i$, with features $\mathbf{x}_i$ and connecting neighboring nodes, $i$ and $j$, via edges $ij$, with features $\boldsymbol{\varepsilon}_{ij}$.

of model iterations, which slow down numerical models over long time horizons. Deep learning provides an opportunity to accelerate this process.

## 2.2 Deep learning

Deep learning obtains non-linear high dimensional representations from data, via multiple levels of abstraction (LeCun et al., 2015). The key building block of deep learning models are neural networks, which comprise linear and non-linear parametric functions. They take an input $\mathbf{x}$ and produce an estimate $\hat{\mathbf{y}}$ of a target representation $\mathbf{y}$ as $\hat{\mathbf{y}} = f(\mathbf{x}; \theta)$, where $\theta$ are the parameters (Zhang et al., 2021). The parameters are estimated to match predicted output with the real output by minimizing a loss function. Then, the validity of the model is assessed by measuring its performance on a set of unseen pairs of data, called the test set.

The most general type of neural network is the multi-layer perceptron (MLP). It is formed by stacking linear models followed by a point-wise non-linearity (e.g., ReLU, $\sigma(x) = \max\{0, x\}$). For MLPs, the number of parameters and the computational cost increase exponentially with the dimensions of the input. This makes them unappealing to large scale high-dimensional data typical of problems with relevant spatio-temporal features such as floods. MLPs are non-inductive: when trained for flood prediction on a certain topography, they cannot be deployed on a different one, thus requiring a complete retraining. To overcome this curse of dimensionality and increase generalizability, models can include inductive biases that constrain their degrees of freedom by reusing parameters and exploiting symmetries in the data (Battaglia et al., 2018; Gama et al., 2020; Villar et al., 2023). For example, convolutional neural networks exploit translational symmetries via filters that share parameters in space (e.g., LeCun et al., 2015; Bronstein et al., 2021). However, CNNs cannot process data defined on irregular meshes, which are common for discretizing topographies with sparse details. Thus, we need a different inductive bias for data on meshes.

Graph neural networks (GNNs) use graphs as an inductive bias to tackle the curse of dimensionality. This bias can be relevant for data represented via networks and meshes, as it allows these models to generalize to unseen graphs, i.e., the same model can be applied to different topographies discretized by different meshes. GNNs work by propagating features defined on the nodes,

based on how they are connected. The propagation rule is then essential in correctly modelling a physical system. However, standard GNNs do not include physics-based rules meaning that the propagation rules may lead to unrealistic results.

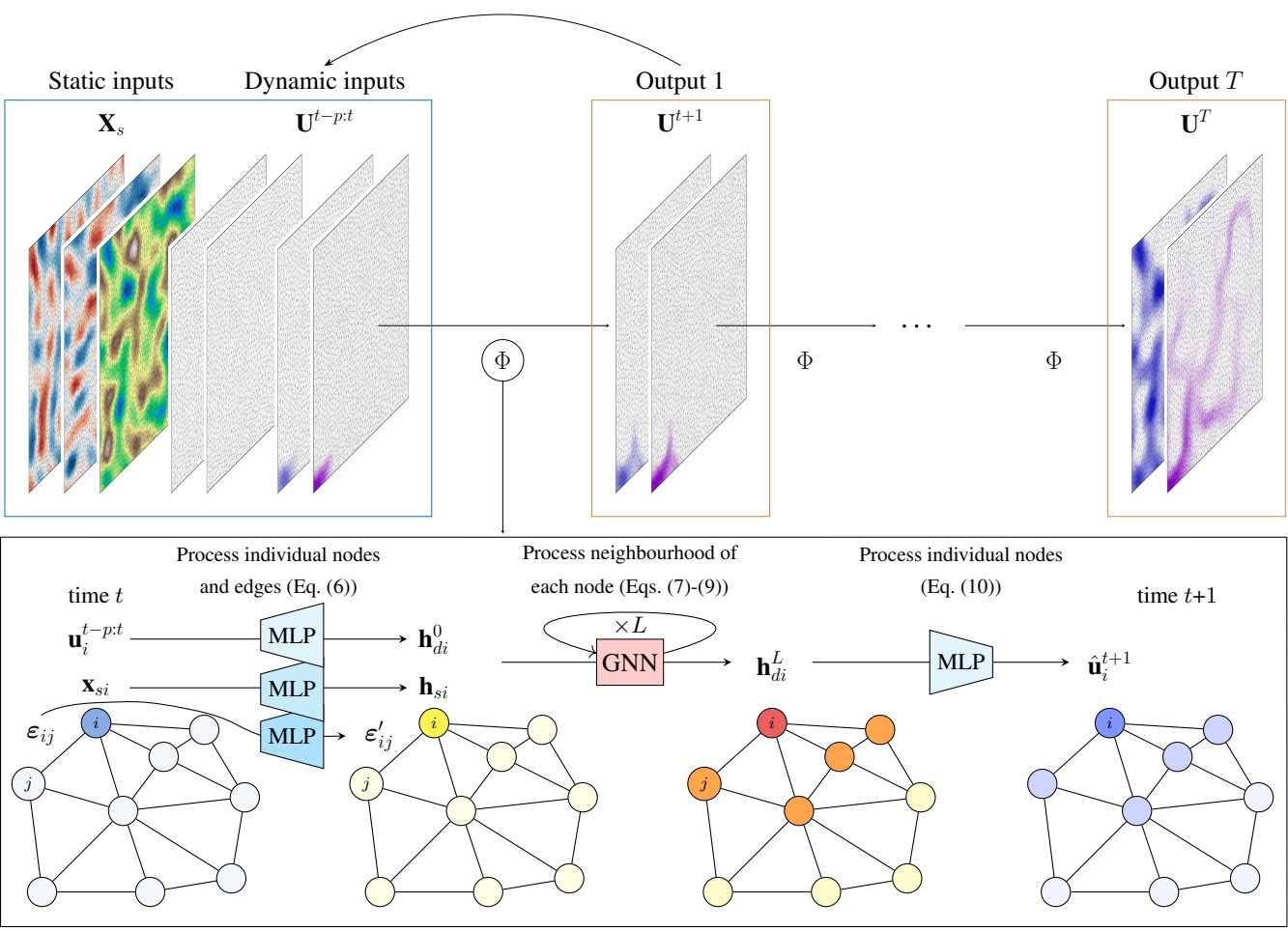

**Figure 2.** Overview of the proposed SWE-GNN model. The model $\Phi$ takes as input the mesh discretization of the static and dynamic input (blue box) and produces an estimate of their evolution in time (orange box). The model is then repeated auto-regressively, i.e., using its predictions as inputs, to determine the spatio-temporal evolution of the flood. The encoder-processor-decoder structure of the SWE-GNN model is shown in the bottom black box. The node inputs $\mathbf{x}_{si}$ and $\mathbf{u}_i^{t-p:t}$ represent static attributes, such as elevation and slopes, and dynamic attributes, representing hydraulic variables, while the edge inputs $\boldsymbol{\varepsilon}_{ij}$ represent the mesh's geometry. The inputs are encoded into higher-dimensional embeddings $\mathbf{h}_{si}$, $\mathbf{h}_{di}^0$ (yellow nodes), and $\boldsymbol{\varepsilon}_{ij}'$ via three separate multi-layer perceptrons, shared across nodes or edges. The embeddings, whose purpose is to increase the inputs' expressivity, are used as input for the $L$ GNN layers. The output of the GNN $\mathbf{h}_{di}^L$ (red and orange nodes) is decoded via another shared multi-layer perceptron and summed to the hydraulic variables at time $t$ $\mathbf{u}_i^t$. The final output $\hat{\mathbf{y}}_i$ (blue nodes) represents the prediction at time $t+1$, i.e., $\hat{\mathbf{u}}_i^{t+1}$.

## 3 Shallow water equations-inspired graph neural network (SWE-GNN)

We develop a graph neural network in which the computations are based on the shallow-water equations. The proposed model takes as input both static and dynamic features that represent the topography of the domain and the hydraulic variables at time $t$, respectively. The outputs are the predicted hydraulic variables at time $t+1$. In the following, we detail the proposed model (Section 3.1) and its inputs and outputs (Section 3.2). Finally, we discuss the training strategy (Section 3.3).

### 3.1 Architecture

SWE-GNN is an encoder-processor-decoder architecture inspired by You et al. (2020) with residual connections, that predicts autoregressively the hydraulic variables at time $t+1$ as

$$\hat{\mathbf{U}}^{t+1} = \mathbf{U}^t + \Phi(\mathbf{X}_s, \mathbf{U}^{t-p:t}, \boldsymbol{\mathcal{E}}), \tag{5}$$

where the output $\hat{\mathbf{U}}^{t+1}$ corresponds to the predicted hydraulic variables at time $t+1$, $\mathbf{U}^t$ are the hydraulic variables at time $t$, $\Phi(\cdot)$ is the GNN-based encoder-processor-decoder model that determines the evolution of the hydraulic variables for a fixed

time step, $\mathbf{X}_s$ are the static node features, $\mathbf{U}^{t-p:t}$ are the dynamic node features, i.e., the hydraulic variables for time steps $t-p$ to $t$, and $\boldsymbol{\mathcal{E}}$ are the edge features that describe the geometry of the mesh. The architecture detailed in the sequel is illustrated in Fig. 2.

**Encoder.** We employ three separate encoders for processing the static node features $\mathbf{X}_s \in \mathbb{R}^{N \times I_{Ns}}$, dynamic node features $\mathbf{X}_d \equiv \mathbf{U}^{t-p:t} \in \mathbb{R}^{N \times O(p+1)}$, and edge features $\boldsymbol{\varepsilon} \in \mathbb{R}^{E \times I_\varepsilon}$, where $I_{Ns}$ is the number of static node features, $O$ the number of

150 hydraulic variables (e.g., $O$=3 if we consider water depth and the $x$ and $y$ components of the unit discharges), $p$ the number of input previous time steps, and $I_\varepsilon$ the number of input edge features. The encoded variables are

$$\mathbf{H}_s = \phi_s(\mathbf{X}_s), \mathbf{H}_d = \phi_d(\mathbf{X}_d), \boldsymbol{\mathcal{E}}' = \phi_\varepsilon(\boldsymbol{\mathcal{E}}), \tag{6}$$

where $\phi_s(\cdot)$ and $\phi_d(\cdot)$ are MLPs shared across all nodes that take an input $\mathbf{X} \in \mathbb{R}^{N \times I}$ and return a node matrix $\mathbf{H} \in \mathbb{R}^{N \times G}$; and $\phi_\varepsilon(\cdot)$ are MLPs shared across all edges that encode the edge features in $\boldsymbol{\mathcal{E}}' \in \mathbb{R}^{E \times G}$. All MLPs have two layers, with hidden

dimension $G$, followed by a *PReLU* activation. The encoders expand the dimensionality of the inputs to allow for higher expressivity, with the hyperparameter $G$ being the dimension of the node embeddings. The $i^{th}$ rows of the node matrices $\mathbf{H}_s$ and $\mathbf{H}_d$ represent the encoded feature vectors associated to node $i$, i.e., $\mathbf{h}_{si}$ and $\mathbf{h}_{di}$, and the $k^{th}$ rows of the edge matrices $\boldsymbol{\mathcal{E}}'$ represents the encoded feature vector associated to edge $k$.

**Processor.** We employed as processor an $L$-layer GNN that takes a high-dimensional representation of the static and dynamic

properties of the system at time $t$, given by the encoders, and produces a spatio-temporally propagated high-dimensional representation of the system's evolution from time $t$ to $t+1$. The propagation rule is based on the shallow water equation. In the SWE, the mass and momentum fluxes, representative of the dynamic features, evolve in space as a function of the source terms, representative of the static and dynamic features. Moreover, water can only propagate from sources of water and the velocity of propagation is influenced by the gradients of the hydraulic variables. Thus, the GNN layer $\ell = 1, \ldots, L-1$ update

reads as

$$\mathbf{s}_{ij}^{(\ell+1)} = \psi\left(\mathbf{h}_{si}, \mathbf{h}_{sj}, \mathbf{h}_{di}^{(\ell)}, \mathbf{h}_{dj}^{(\ell)}, \varepsilon'_{ij}\right) \odot \left(\mathbf{h}_{dj}^{(\ell)} - \mathbf{h}_{di}^{(\ell)}\right), \tag{7}$$

$$\mathbf{h}_{di}^{(\ell+1)} = \mathbf{h}_{di}^{(\ell)} + \sum_{j \in \mathcal{N}_i} \mathbf{s}_{ij}^{(\ell+1)} \mathbf{W}^{(\ell+1)}, \tag{8}$$

where $\psi(\cdot) : \mathbb{R}^{5G} \to \mathbb{R}^G$ is an MLP with two layers, with hidden dimension $2G$, followed by a *PReLU* activation function, $\odot$ is the Hadamard (element-wise) product, and $\mathbf{W}^{(\ell)} \in \mathbb{R}^{G \times G}$ are parameter matrices. The term $\mathbf{h}_{dj}^{(\ell)} - \mathbf{h}_{di}^{(\ell)}$ represents the

gradient of the hydraulic variables and enforces water-related variables $\mathbf{h}_d$ to propagate only if at least one of the interfacing node features is non-zero, i.e., has water. The function $\psi(\cdot)$, instead, incorporates both static and dynamic inputs and provides an estimate of the source terms acting on the nodes. Thus, vector $\mathbf{s}_{ij}$ represents the fluxes exchanged across neighbouring cells and their linear combination is used as in eq. (3) to determine the hydraulic variables' variation for a given cell. In this way, eq. (7) resembles how fluxes are evaluated at the cell's interface in the numerical model, i.e., $\delta \mathbf{F}(\mathbf{u})_{ij} = \tilde{\mathbf{J}}_{ij}(\mathbf{u}_j - \mathbf{u}_i)$, which

enforces conservation across interface discontinuities (Martínez-Aranda et al., 2022). Based on this formulation, $\mathbf{s}_{ij}$ can also be interpreted as approximate Riemann solver (Toro, 2013), where the Riemann problem at the boundary between computational cells is approximated by function $\psi(\cdot)$, in place of equations (e.g., Roe, 1981). To reduce model instabilities, the output of $\psi(\cdot)$ is normalized along its embedding dimension, i.e., it is divided by its norm $\|\psi(\cdot)\|$. This procedure is similar to other graph normalization techniques that improve training stability (Chen et al., 2022). The contribution of each layer is linearly

multiplied by $\mathbf{W}^{(\ell)}$ (Eq. (7)). From a numerical perspective, this is analogous to an $L$-order multi-time-step scheme, with $L$ being the number of layers, where the weights are learned instead of being assigned (e.g., Dormand and Prince, 1980).

The GNN's output represents an embedding of the predicted hydraulic variables at time $t+1$ for a fixed time step $\Delta t$. Instead of enforcing stability by limiting $\Delta t$, as it is done in numerical models, we can obtain the same result by considering a larger portion of space, which results in increasing $\Delta x$ (cfr. eq. (4)). This effect can be achieved by stacking multiple GNN layers, as

each layer will increase the propagation space, also called neighborhood size. The number of GNN layers is then correlated to the space covered by the flood for a given temporal resolution. We can then write the full processor for the $L$ GNN layers as

$$\mathbf{h}_{di}^{(0)} = \mathbf{h}_{di}\mathbf{W}^{(0)},$$

$$\mathbf{s}_{ij}^{(\ell+1)} = \psi\left(\mathbf{h}_{si}, \mathbf{h}_{sj}, \mathbf{h}_{di}^{(\ell)}, \mathbf{h}_{dj}^{(\ell)}, \varepsilon'_{ij}\right) \odot \left(\mathbf{h}_{dj}^{(\ell)} - \mathbf{h}_{di}^{(\ell)}\right),$$

$$\mathbf{h}_{di}^{(\ell+1)} = \mathbf{h}_{di}^{(\ell)} + \sum_{j \in \mathcal{N}_i} \mathbf{s}_{ij}^{(\ell+1)} \mathbf{W}^{(\ell+1)},$$

$$\mathbf{h}_{di}^{(L)} = \sigma\left(\mathbf{h}_{di}^{(L-1)} + \sum_{j \in \mathcal{N}_i} \mathbf{s}_{ij}^{(L)} \mathbf{W}^{(L)}\right), \tag{9}$$

where we employ a Tanh activation function $\sigma(\cdot)$ at the output of the $L^{th}$ layer to limit numerical instabilities resulting in

exploding values. The embedding of the static node features $\mathbf{h}_{si}$ and of the edge features $\varepsilon'_{ij}$ do not change across layers, as the topography and discretization of the domain do not change in time.

**Decoder.** Symmetrically to the encoder, the decoder is composed of an MLP $\varphi(\cdot)$, shared across all the nodes, that takes as input the output of the processor $\mathbf{H}_d^{(L)} \in \mathbb{R}^{N \times G}$ and updates the hydraulic variables at the next time step, i.e., $\hat{\mathbf{U}}^{t+1} \in \mathbb{R}^{N \times O}$,

via residual connections, as

$$\hat{\mathbf{U}}^{t+1} = \mathbf{U}^t + \varphi\left(\mathbf{H}_d^{(L)}\right). \tag{10}$$

The MLP $\varphi(\cdot)$ has two layers, with hidden dimension $G$, followed by a *PReLU* activation. Both the MLPs in the dynamic encoder and the decoder do not have the bias terms as this would result in adding non-zero values in correspondence of dry areas that would cause water to originate from any node.

## 3.2 Inputs and outputs

We define input features on the nodes and edges based on the SWE terms (cfr. eq. (2)). We divide node features into a static component that represents fixed spatial attributes and a dynamic component that represents the hydraulic variables.

**Static node features** are defined as

$$\mathbf{x}_{si} = (a_i, e_i, \mathbf{s}_{0i}, m_i, w_i^t) \tag{11}$$

where $a_i$ is the area of the $i^{th}$ finite volume cell, its elevation $e_i$, its slopes in the $x$ and $y$ directions $\mathbf{s}_{0i}$, and its Manning coefficient $m_i$. We also included the water level at time $t$, $w_i^t$, given by the sum of elevation and water depth at time $t$, as node inputs, since it determines the water gradient (Liang and Marche, 2009). The reason why we include $w_i^t$ in the static attributes instead of the dynamic ones is that this feature can be non-zero also without water, due to the elevation term, and would thus result in the same issue mentioned for the dynamic encoder and decoder.

**Dynamic node features** are defined as

$$\mathbf{x}_{di} = \mathbf{u}_i^{t-p:t} = (\mathbf{u}_i^{t-p}, ..., \mathbf{u}_i^{t-1}, \mathbf{u}_i^t),$$
$$\mathbf{u}_i^t = (h_i^t, |q|_i^t) \tag{12}$$

where $\mathbf{u}_i^t$ are the hydraulic variables at time step $t$ and $\mathbf{u}_i^{t-p:t}$ are the hydraulic variables up to $p$ previous time steps, to leverage the information of past data and provide a temporal bias to the inputs. Contrarily to the definition of the hydraulic variables as in Eq. (2), we selected the modulus of the unit discharge $|q|$ as a metric of flood intensity in place of its $x$ and $y$ components to avoid mixing scalar and vector components and because, for practical implications, such as damage estimation, the flow direction is less relevant than its absolute value (e.g., Kreibich et al., 2009).

**Edge features** are defined as

$$\varepsilon_{ij} = (\mathbf{n}_{ij}, l_{ij}), \tag{13}$$

where $\mathbf{n}_{ij}$ is the outward unit normal vector and $l_{ij}$ is the cell's sides length. Thus, the edge features represent the geometrical properties of the mesh. We excluded the fluxes $\mathbf{F}_{ij}$ as additional features as they depend on the hydraulic variables $\mathbf{u}_i$ and $\mathbf{u}_j$, which are already included in the dynamic node features.

**Outputs**. The model outputs are the estimated water depth and unit discharge at time $t+1$, i.e., $\hat{\mathbf{u}}_i^{t+1} = \left(\hat{h}_i^{t+1}, |\hat{q}|_i^{t+1}\right)$, resulting in an output dimension $O = 2$. The outputs are used to update the input dynamic node features $\mathbf{x}_{di}$ for the following time step, as exemplified in Fig. 3. The same applies for the water level in the static attributes, i.e., $w_i^{t+1} = e_i + \hat{h}_i^{t+1}$.

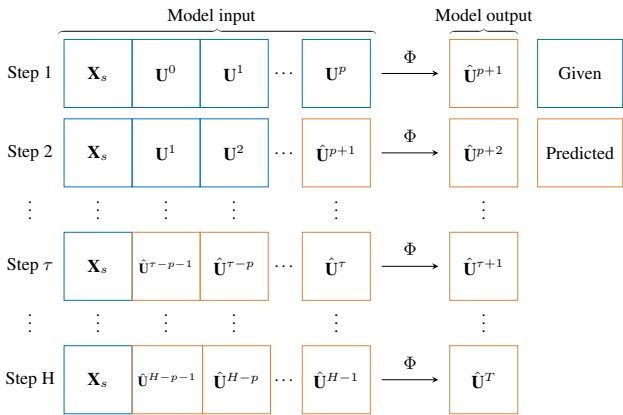

**Figure 3.** Example of auto-regressive prediction for $p$ input previous time steps and $H$ predicted steps ahead. The prediction at time $\tau$ are used as new inputs to predict the following time step and so on. The loss and the metrics are evaluated as the average over all steps $H$.

## 3.3 Training strategy

The model learns from input-output data pairs. To stabilize the output of the SWE-GNN over time, we employ a multi-step-ahead loss function $\mathcal{L}$, that measures the accumulated error for multiple consecutive time steps, i.e.,

$$\mathcal{L} = \frac{1}{HO} \sum_{\tau=1}^{H} \sum_{o=1}^{O} \gamma_o \|\hat{\mathbf{u}}_o^{t+\tau} - \mathbf{u}_o^{t+\tau}\|_2, \tag{14}$$

where $\mathbf{u}_o^{t+\tau} \in \mathbb{R}^N$ are the hydraulic variables over the whole graph at time $t+\tau$, $H$ is the prediction horizon, i.e., the number of consecutive time instants, and $\gamma_o$ are coefficients used to weight the influence of each variable to the loss. For each time step $\tau$, we evaluate the model's prediction $\hat{\mathbf{u}}^{t+\tau}$ and then use the prediction recursively as part of the new dynamic node input (see Fig. 3). We repeat this process for a number of time steps $H$ and calculate the root mean squared error (RMSE) loss as the average over all steps. In this way, the model learns to correct its own predictions while also learning to predict a correct output, given a slightly wrong prediction, hence improving its robustness. After $p+1$ prediction steps, the inputs of the model are given exclusively by its predictions. During training, we limit the prediction horizon $H$ instead of using the full temporal sequence due to memory constraints, since the back-propagation gradients must be stored for each time step.

To improve the training speed and stability, we also employed a curriculum learning strategy (Algorithm 1). This consists in progressively increasing the prediction horizon in eq. (14) every fixed number of epochs up to $H$. The idea is to first learn the one- or few-steps-ahead predictions to fit the short-term predictions and then increase the number of steps ahead to stabilize the predictions (Wang et al., 2021).

**Algorithm 1** Curriculum learning strategy

---

**Initialize:**

    $H = 1$

    $CurriculumSteps = 15$

    $\gamma_1 = 1$ (Water depth $h$)

    $\gamma_2 = 3$ (Unit discharge $q$)

**for** epoch = 1 to MaxEpochs **do**

    $\hat{\mathbf{U}}^{t+1} = \mathbf{U}^t + \Phi(\mathbf{X}_s, \mathbf{U}^{t-p:t}, \boldsymbol{\mathcal{E}})$

    $\mathcal{L} = \frac{1}{HO} \sum_{\tau=1}^{H} \sum_{o=1}^{O} \gamma_o \|\hat{\mathbf{u}}_o^{t+\tau} - \mathbf{u}_o^{t+\tau}\|_2,$

    Update the parameters

    **if** epoch > CurriculumSteps*H **then**

        $H = H + 1$

    **end if**

**end for**

---

## 4 Experimental setup

### 4.1 Dataset generation

We considered 130 numerical simulations of dike-breach floods ran on randomly-generated topographies over two squared domains of sizes $6.4 \times 6.4 km^2$ and $12.8 \times 12.8 km^2$ representative of flood-prone polder areas.

We generated random digital elevation models using the Perlin noise generator (Perlin, 2002) as its ups and downs reflect plausible topographies. We opted for this methodology, instead of manually selecting terrain patches, to automatize the generation process, thus allowing for an indefinite amount of randomized and unbiased training and testing samples.

We employed a high-fidelity numerical solver, Delft3D-FM, which solves the full shallow water equations using an implicit scheme on staggered grids and adaptive time steps (Deltares, 2022). We used a dry bed as the initial condition and a constant input discharge of $50 m^3/s$ as the boundary condition, equal to the maximum dike-breach discharge. We employed a single

**Table 1.** Summary of the datasets employed for training (TR), validation (VA), and testing (TE). The uncertainty accounts for the variability across the different simulations in each dataset.

| Dataset and use | Number of simulations | Size $(km^2)$ | Random breach location | Simulation duration $(h)$ | Execution time of numerical model $(s)$ |
|---|---|---|---|---|---|
| 1 (TR,VA,TE) | 100 | $6.4 \times 6.4$ | No | 48 | $29.5 \pm 9.1$ |
| 2 (TE) | 20 | $6.4 \times 6.4$ | Yes | 48 | $32.5 \pm 5.1$ |
| 3 (TE) | 10 | $12.8 \times 12.8$ | Yes | 120 | $185.5 \pm 29.9$ |

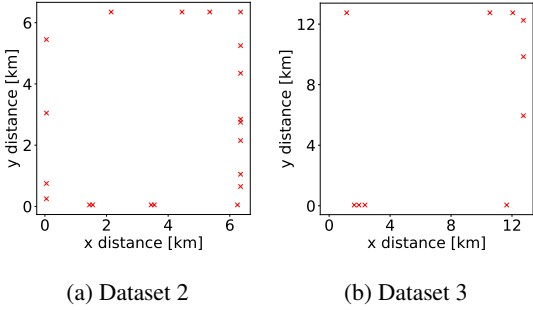

(a) Dataset 2       (b) Dataset 3

**Figure 4.** Distribution of the breach locations (red crosses) for datasets 2 and 3.

boundary condition value for all simulations as our focus is to show generalizability over different topographies and breach locations. The simulation output is a set of temporally-consecutive flood maps, with a temporal resolution of 30 minutes.

We created three datasets with different area sizes and breach locations as summarized in Table 1. We selected a rectangular domain discretized by regular meshes, to allow for a fairer comparison with other models that cannot work with meshes or cannot incorporate edge attributes. Furthermore, we considered a constant roughness coefficient $m_i$ for all simulations, meaning that we use the terrain elevation and the slopes in the $x$ and $y$ directions as static node inputs.

1. The first dataset consists of 100 DEMs over a squared domain of $64 \times 64$ grids of length $100m$ and a simulation time of 48 hours. This dataset is used for training, validation, and testing. We used a fixed testing set of 20 simulations while the remaining 80 simulations are used for training (60) and validation (20).

2. The second dataset consists of 20 DEMs over a squared domain of $64 \times 64$ grids of length of $100m$ and a simulation time of 48 hours. The breach location changes randomly across the border with a constant discharge of $50m^3/s$ (Fig. 4a). This dataset is used to test the generalizability of the model to unseen domains and breach locations.

3. The third dataset consists of 10 DEMs over a squared domain of $128 \times 128$ grids of length of $100m$. The boundary conditions are the same as for the second dataset. Since the domain area is four times larger, the total simulation time is 120 hours, to allow for the flood to cover larger parts of the domain. This dataset is used to test the generalizability of the model to larger unseen domains, unseen breach locations, and longer time horizons.

Unless otherwise mentioned, we selected a temporal resolution of $\Delta t$=1$h$, as a trade-off between detail and speed. When the beginning of the flood is relevant (e.g., for real-time forecasts) higher temporal resolutions are better. Contrarily, if the the final flood state is relevant, lower temporal resolutions may be better.

## 4.2 Training setup

We trained all models via the Adam optimization algorithm (Kingma and Ba, 2014). We employed a varying learning rate with 0.005 as starting value and a fixed step decay of 90% every 7 epochs. The training was carried out for 150 epochs with early

stopping. We used a maximum prediction horizon $H = 8$ steps ahead during training as a trade-off between model stability and training time, as later highlighted in Section 5.4. There is no normalization pre-processing step and, thus, the values of water depth and unit discharge differ in magnitude by a factor of 10. Since for application purposes discharge is less relevant than water depth (Kreibich et al., 2009), we weighted the discharge term by a factor of $\gamma_2 = 3$ (cfr. eq. (14)), while leaving the weight factor for water depths as $\gamma_1 = 1$. Finally, we used one previous time step as input, i.e., $\mathbf{X}_d = (\mathbf{U}^{t=0}, \mathbf{U}^{t=1})$, where the solution at time $t = 0$ corresponds to dry bed conditions.

We trained all models using the Pytorch (Version 3.10.8) (Paszke et al., 2019) and Pytorch Geometric (Version 2.2) (Fey and Lenssen, 2019). In terms of hardware, we employed an Nvidia Tesla V100S-PCIE-32GB for training and deployment (Delft High Performance Computing Centre , DHPC), and an Intel(R) Core(TM) i7-8665U @1.9 GHz CPU for deployment and for the execution of the numerical model. We run the models on both GPUs and CPUs to allow for a fair comparison with the numerical models.

## 4.3 Metrics

We evaluated the performance using the multi-step-ahead RMSE (eq. (14)) over the whole simulation. However, for testing, we calculated the RMSE for each hydraulic variable $o$ independently as:

$$RMSE_o = \frac{1}{H} \sum_{\tau=1}^{H} \|\hat{\mathbf{u}}_o^\tau - \mathbf{u}_o^\tau\|_2, \tag{15}$$

Analogously, we evaluated the mean average error (MAE) for each hydraulic variable $o$ over the whole simulation as:

$$MAE_o = \frac{1}{H} \sum_{\tau=1}^{H} \|\hat{\mathbf{u}}_o^\tau - \mathbf{u}_o^\tau\|_1, \tag{16}$$

The prediction horizon $H$ depends on the total simulation time and temporal resolution, e.g., predicting 24 hours with a temporal resolution of 30 min results in $H = 48$ steps ahead. We also measured the spatio-temporal error distribution of the water depth using the critical success index (CSI) for threshold values of 0.05 m and 0.3 m, as in Löwe et al. (2021). The CSI measures the spatial accuracy of detecting a certain class (e.g., flood or no-flood) and, for a given threshold, it is evaluated as

$$CSI = \frac{TP}{TP + FP + FN} \tag{17}$$

where TP are the true positives, i.e., number of cells where both model and simulations predict flood, FP are the false positives, i.e., number of cells where the model wrongly predicts flood, and FN are the false negatives, i.e., number of cells where the model does not recognize a flooded area. We selected this measure as it discards true negatives, i.e., when both model and simulation predict no flood, as this condition is over-represented, especially for the initial time steps. Thus, including true negatives may give an overconfident performance estimate. We measured the computational speed-up as the ratio between the computational time required by the numerical model and the inference time of the deep learning model. Both times refer to the execution of the complete flood simulation but do not include the time required to simulate the initial time steps.

## 5 Numerical Results

### 5.1 Comparison with other deep learning models

The proposed SWE-GNN model is compared with other deep learning methods including:

- CNN: encoder-decoder convolutional neural network, based on U-NET (Ronneberger et al., 2015). The CNN considers the node feature matrix $\mathbf{X}$ reshaped as a tensor $\mathcal{X} \in \mathbb{R}^{g \times g \times I_N}$, where $g$ is the number of grid cells, i.e., 64 for datasets 1 and 2 and 128 for dataset 3, and $I_N$ is the number of static and dynamic features. This baseline is used to highlight the advantages of the mesh dual graph as an inductive bias in place of an image;

- GAT: graph attention network (Velickovic et al., 2017). The weights in the propagation rule are learned, considering an attention-based weighting. This baseline is considered to show the influence of learning the propagation rule with an attention mechanism. For more details see Appendix A;

- GCN: graph convolutional neural network (Defferrard et al., 2016). This baseline is considered to show the influence of not learning the edge propagation rule, in place of learning it. For more details see Appendix A;

- SWE-GNN$_{ng}$: SWE-GNN without the gradient term $\mathbf{x}_{dj} - \mathbf{x}_{di}$. This is used to show the importance of the gradient term in the graph propagation rule.

We evaluated also MLP-based models, but their performance was too poor and we do not report it. All models consider the same node features inputs $\mathbf{X} = (\mathbf{X}_s, \mathbf{X}_d)$, produce the same output $\hat{\mathbf{Y}} = \mathbf{U}^{t+1}$, produce extended simulations by using the predictions as input (as in Fig. 3), and use the same training strategy with the multi-step-ahead loss and curriculum learning. For the GNN-based models, we replaced the GNN in the processor, while keeping the encoder-decoder structure as in Fig. 2. We conducted a thorough hyperparameter search for all models, and we selected the one with the best validation loss. For the CNN architecture,

**Table 2.** Performance of the deep learning models over the test dataset 1. The provided uncertainty estimates account for the variability across the different simulations in the dataset. **Bold** results indicate the best performances, considering a statistical significance with a $p$-value of 0.05.

| DL model | RMSE | | MAE | | CSI$_\tau$ [%] | |
|---|---|---|---|---|---|---|
| | h $(m)$ $[10^{-2}]$ | $\lvert q \rvert$ $(m^2/s)$ $[10^{-2}]$ | h $(m)$ $[10^{-2}]$ | $\lvert q \rvert$ $(m^2/s)$ $[10^{-2}]$ | $\tau$=0.05 $m$ | $\tau$=0.3 $m$ |
| CNN | **10.97 ± 5.11** | **1.33 ± 0.57** | **3.87 ± 1.29** | **0.42 ± 0.13** | **75.64 ± 9.40** | **73.42 ± 9.26** |
| GAT | 25.78 ± 7.23 | 1.96 ± 0.61 | 9.27 ± 0.73 | 5.78 ± 0.11 | 34.50 ± 10.91 | 27.07 ± 8.63 |
| GCN | 16.49 ± 6.91 | 1.65 ± 0.55 | 6.05 ± 1.62 | 0.57 ± 0.11 | 61.14 ± 13.34 | 58.89 ± 11.90 |
| SWE-GNN_$ng$ | 16.24 ± 6.65 | 1.71 ± 0.66 | 6.10 ± 1.56 | 0.63 ± 0.07 | 58.61 ± 11.97 | 57.91 ± 12.62 |
| SWE-GNN | **11.15 ± 5.11** | **1.22 ± 0.42** | **3.93 ± 1.63** | **0.37 ± 0.10** | **75.85 ± 9.30** | **73.44 ± 9.28** |

the best model has three down- and up-scaling blocks, with 64 filters in the first encoding block. Interestingly, we achieved good results only when employing batch normalization layers, *PReLU* as an activation function, and no residual connections. All other standard combinations resulted in poor performances, which we did not report as they are outside the scope of the paper. For the GNN-based architectures, all hyperparameter searches resulted in similar best configurations, i.e., $L = 8$ GNN layers and an embedding size $G$=64.

In Table 2, we report the testing RMSE and MAE for water depth and discharges, and the CSI scores for all models. The proposed SWE-GNN model and the U-NET-based CNN perform consistently better than all other models, with no statistically significant difference in performance according to the Kolmogorow-Smirnov test (p-value less than 0.05). The CNN performs similar to the SWE-GNN because the computations on a regular grid are similar to those of a GNN. Nonetheless, there are valuable differences between the two models. First, SWE-GNN is by definition more physically explainable as water can only propagate from wet cells to neighboring cells, while in the CNN there is no such physical constraint, as exemplified by Fig. 5b. Second, as emphasized in the following section, the SWE-GNN results in improved generalization abilities. Moreover, contrarily to CNNs, GNNs can also work with irregular meshes. Regarding the other GNN-based models, we noticed that the GAT model had the worse performance, indicating that the propagation rule cannot be learned efficiently via attention mechanisms. Moreover, the GCN and the SWE-GNN$_{ng}$ achieved comparable results meaning that the gradient term gives a relevant contribution to the model as its removal results in a substantial loss in performance. We expected this behavior as, without this term, there is no computational constraint to how water propagates.

## 5.2 Generalization to other breach locations and larger areas

We further tested the already trained models on datasets 2 and 3, with unseen topographies, unseen breach locations, larger domain sizes, and longer simulation times, as described in Table 1. In the following, we omit the other GNN-based models, since their performance was poorer, as highlighted in Table 2.

**Table 3.** Performance of the deep learning models over the test datasets 2 and 3, respectively composed of unseen domains with unseen breach locations and unseen domains four times bigger than the training ones, also with unseen breach locations. The provided uncertainty estimates account for the variability across different simulations. **Bold** results indicate the best performances, considering a statistical significance with a $p$-value of 0.05.

| Test dataset | DL model | RMSE | | MAE | | CSI$_\tau$ [%] | |
|---|---|---|---|---|---|---|---|
| | | h $(m)$ $[10^{-2}]$ | $|q|$ $(m^2/s)$ $[10^{-2}]$ | h $(m)$ $[10^{-2}]$ | $|q|$ $(m^2/s)$ $[10^{-2}]$ | $\tau$=0.05 $m$ | $\tau$=0.3 $m$ |
| 2 | CNN | $15.74 \pm 7.00$ | $1.69 \pm 0.47$ | $6.50 \pm 2.37$ | $0.54 \pm 0.13$ | $51.90 \pm 20.25$ | $47.82 \pm 18.42$ |
| | SWE-GNN | $\mathbf{11.11 \pm 4.65}$ | $\mathbf{1.31 \pm 0.44}$ | $\mathbf{4.84 \pm 1.87}$ | $\mathbf{0.48 \pm 0.13}$ | $\mathbf{73.62 \pm 8.04}$ | $\mathbf{68.46 \pm 7.13}$ |
| 3 | CNN | $16.86 \pm 3.12$ | $\mathbf{1.21 \pm 0.16}$ | $6.07 \pm 1.77$ | $\mathbf{0.36 \pm 0.10}$ | $42.16 \pm 15.63$ | $40.92 \pm 15.96$ |
| | SWE-GNN | $\mathbf{11.38 \pm 3.95}$ | $\mathbf{1.12 \pm 0.30}$ | $\mathbf{3.77 \pm 1.98}$ | $\mathbf{0.31 \pm 0.12}$ | $\mathbf{68.53 \pm 10.18}$ | $\mathbf{64.53 \pm 11.20}$ |

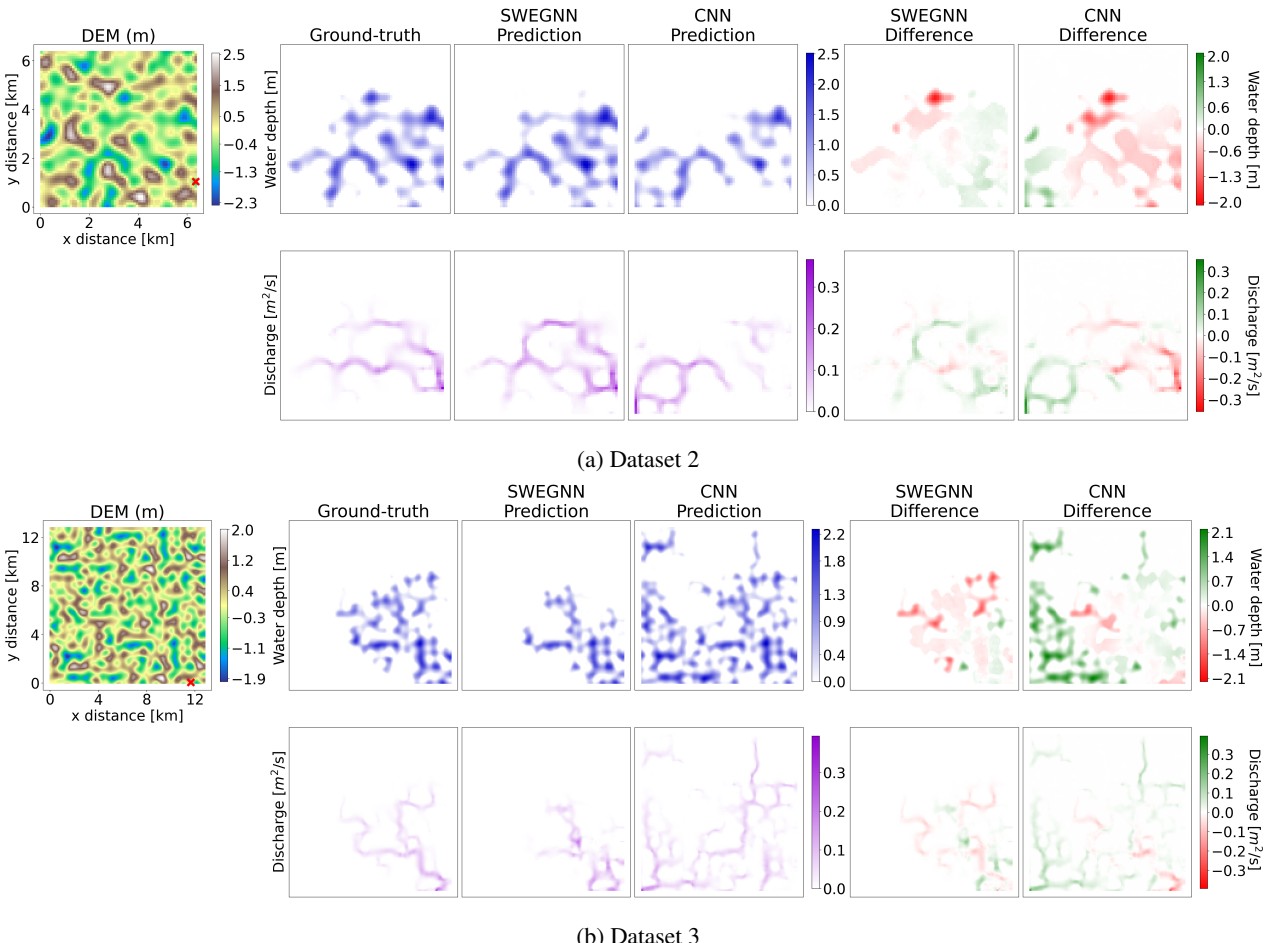

**Figure 5.** Comparison of the proposed SWEGNN model against the CNN, for two examples in test datasets 2 (a) and 3 (b). In each panel, the top left image represents the digital elevation model (DEM), along with a red cross in correspondence of the breach location. The following blocks represent, respectively, the ground-truth numerical results, the SWE-GNN predictions, and the CNN predictions for water depth and unit discharges, at the last time instant of the simulation (i.e., $48h$ for dataset 2 and $120h$ for dataset 3).

Table 3 shows that all metrics remain comparable across the various datasets for SWE-GNN, with test MAE of approximately $0.04m$ for water depth and $0.004m^2/s$ for unit discharges, indicating that the model has learned the dynamics of the problems. The speed-up on GPU of SWE-GNN over dataset 3 further increased, with respect to the smaller areas of dataset 1 and 2, reaching values twice as higher, i.e., ranging from 100 to 600 times faster than the numerical model on the GPU. We attribute this to the deep learning models' scalability and better exploitation of the hardware for larger graphs.

In Figure 5, we see two examples of SWE-GNN and CNN on the test datasets 2 and 3. The SWE-GNN model predicts better the flood evolution over time for unseen breach locations, even in bigger and unseen topographies, thanks to its hydraulic-based approach. On the other hand, the CNN strongly over- or under-predicts the flood extents, unless the breach location is

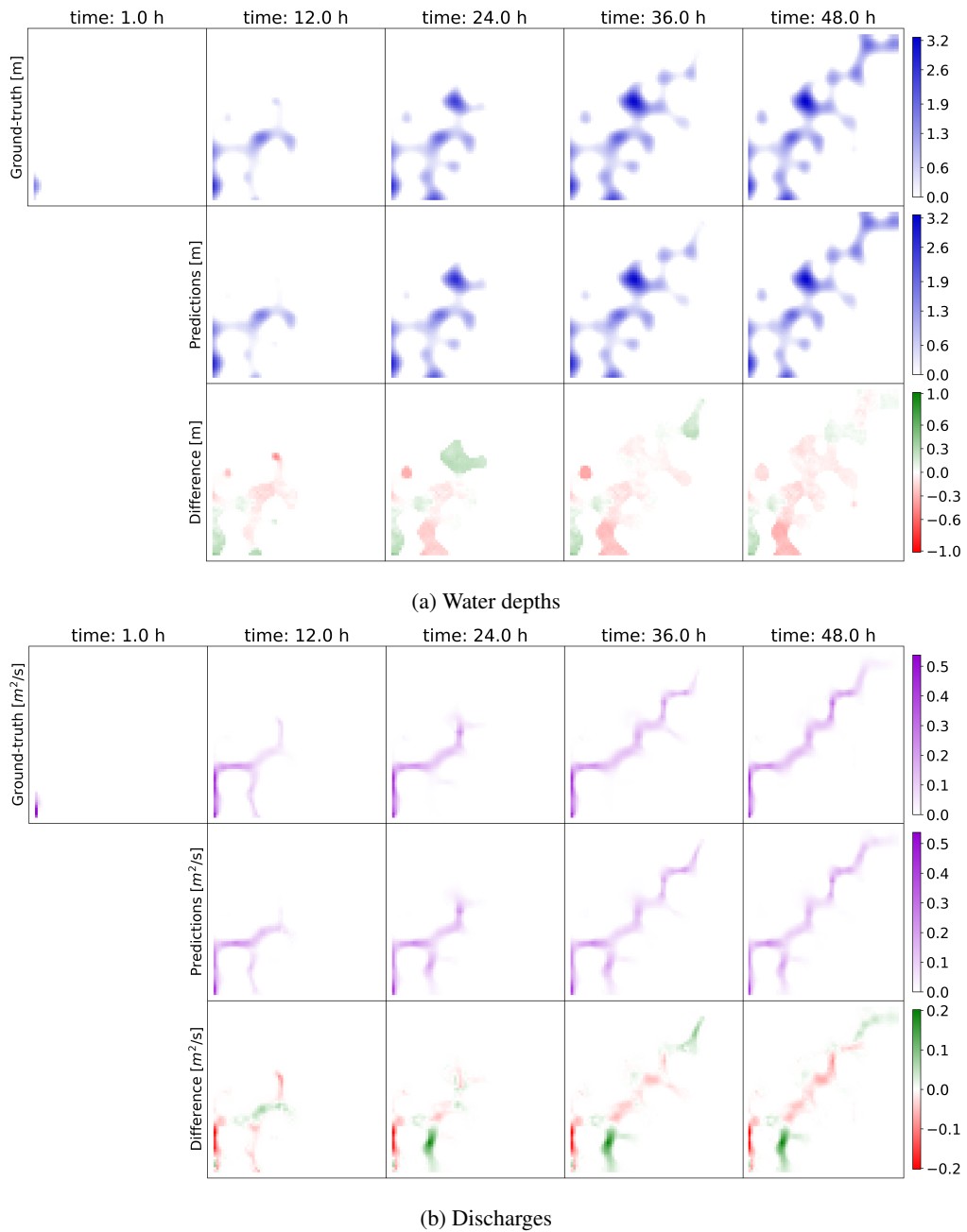

**Figure 6.** SWE-GNN model's predictions for water depth (a) and discharges (b). The results are displayed over time for a test topography in dataset 1, comparing the ground-truth output of the numerical simulation (top row) with the predictions (middle row). The difference (bottom row) is evaluated as the predicted value minus the ground-truth one; thus, positive values correspond to model over-predictions while negative values correspond to under-predictions. The legends refer to the maximum values throughout the whole simulation. The top left panel in both sub-figures represents the initial hydraulic conditions given as input to the DL model, along with the dry bed conditions at time $t = 0$.

close to that of the training dataset, indicating that it lacks the correct inductive bias to generalize floods. For both models, the predictions remain stable even for time horizons 2.5 times longer than those in training.

## 5.3   SWE-GNN model analysis

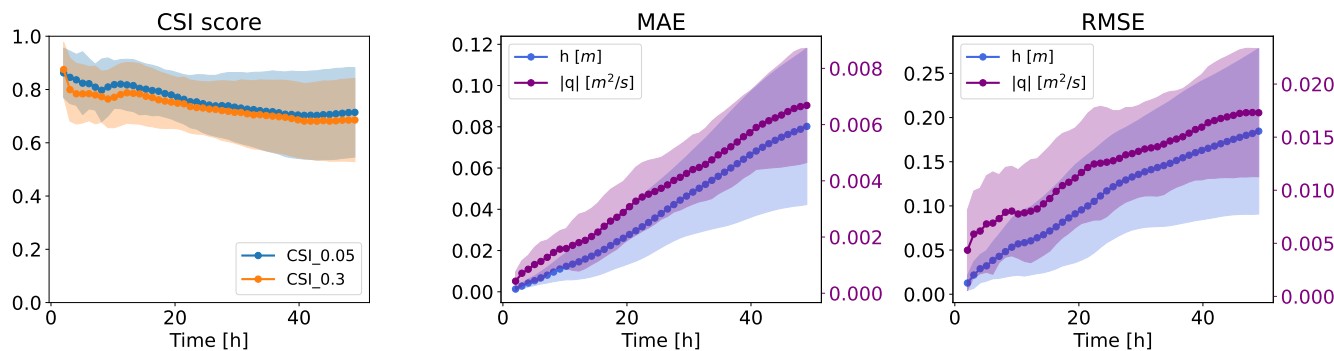

**Figure 7.** Temporal evolution of CSI scores, MAE, and RMSE for test dataset 1. The confidence bands refer to one standard deviation from the mean.

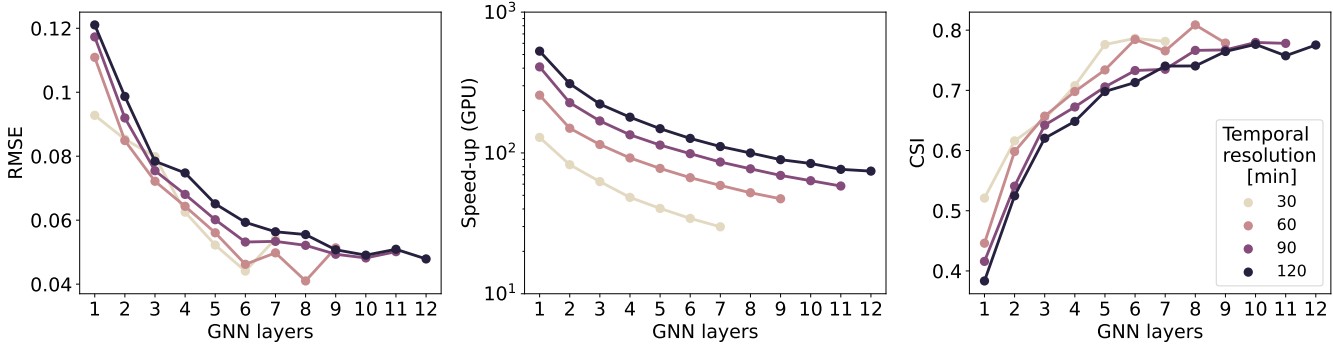

**Figure 8.** Relationship between the number of GNN layers and different temporal resolutions, in terms of validation RMSE and validation CSI. As the temporal resolution decreases and, conversely, as the time step increases, the optimal number of GNN layers, in terms of desired performance level, increases.

Over the entire test part of dataset 1, the model achieves an MAE of $0.04m$ for water depth and $0.004m^2/s$ for unit discharges, with respect to maximum water depths and unit discharges respectively of $2.88m$ and $0.55m^2/s$, and average water

depths and unit discharges of $0.62m$ and $0.037m^2/s$.

We illustrate the spatio-temporal performance of the model on a test sample in Figure 6. Water depth and discharges evolve accurately over time, overall matching the ground-truth numerical results. The errors are related to small over- or underpredictions, a few incorrect flow routes, and lags in the predictions resulting in delays or anticipations that are corrected

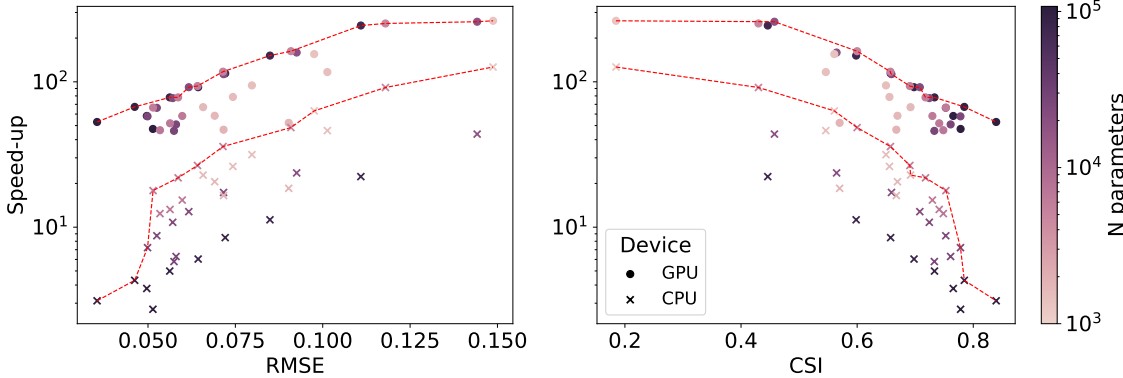

**Figure 9.** Pareto fronts (red-dotted lines) in terms of speed-ups, RMSE, and CSI for varying number of parameters, both for CPU and GPU, for a temporal resolution $\Delta t$=1$h$.

by the successive model iterations. In particular, the model struggles to represent discharges in correspondence of ponding phenomena, i.e., when an area gets filled with water and then forms a temporary lake, as exemplified in the bottom-left part of the domain in Figure 6b. This is because of the lower contribution of the discharges to the training loss. Nonetheless, the error does not propagate over time, thanks to the multi-step-ahead loss employed during training. In fact, the model updates the solution for the entire domain at each time step. Consequently, it exploits information on newly flooded neighborhoods to recompute better values for the cells that were flooded before.

We also observe the average performance of the different metrics over time, for the whole test dataset 1, in Figure 7. The CSI is consistently high throughout the whole simulation, indicating that the model correctly predicts where water is located in space and time. On the other hand, both MAE and RMSE increase over time. This is partially due to the evaluation of both metrics via a spatial average, which implies that in the first time steps, where the domain is mostly dry, the error will naturally be lower. Nonetheless, the errors increase linearly or sub-linearly, implying that they are not prone to explode exponentially.

Next, we analyzed the relationship between the number of GNN layers and the temporal resolution, to validate the hypothesis that the number of layers is correlated with the time steps. Following the CFL condition, we can expand the computational domain by increasing the number of GNN layers in the model instead of decreasing the time steps. We considered several models with an increasing number of GNN layers targeting temporal resolutions $\Delta t = 30, 60, 90, 120 min$. Figure 8 shows that lower temporal resolutions (e.g., $120min$) require more GNN layers to reach the same performance as that of higher temporal resolutions (e.g., $30min$). One reason why the number of layers does not increase linearly with the temporal resolution may be that the weighting matrices $\mathbf{W}_\ell$ (cfr. eq. (7)) improve the expressive power of each layer, leading to fewer layers than needed otherwise.

Finally, we explored different model complexity combinations, expressed by the number of GNN layers and the latent space size, to determine a Pareto-front for validation loss and speed-up, which results in a trade-off between fast and accurate models. Figure 9 shows that increasing the complexity reduces both errors and speed-ups while improving the CSI, as expected. While

for the GPU the number of hidden features does not influence the speed-up, the performance on the CPU depends much more on it, with bigger models being slower, implying different trade-off criteria for deployment.

## 5.4 Sensitivity analysis on the training strategy

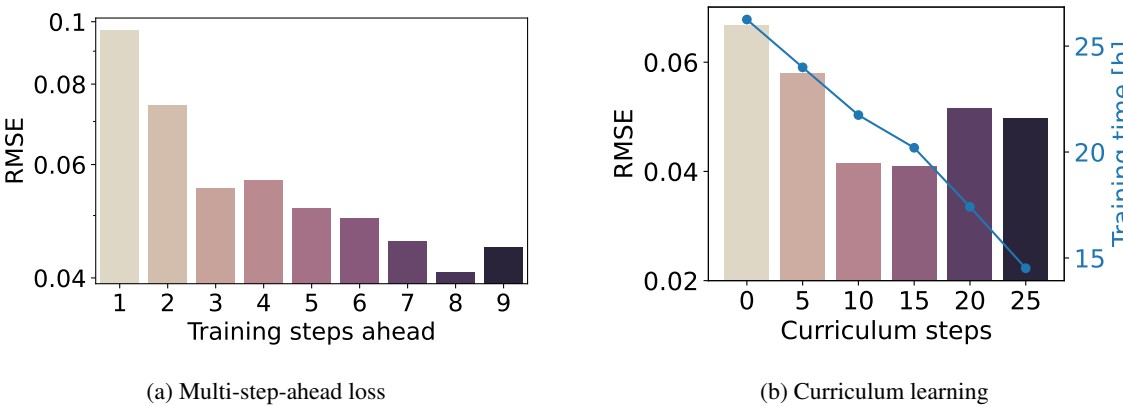

(a) Multi-step-ahead loss          (b) Curriculum learning

**Figure 10.** Influence of: (a) the number of training steps ahead on the validation RMSE and (b) the update interval in the curriculum learning.

Finally, we performed a sensitivity analysis on the role of the multi-step-ahead function (cfr. eq. (14)) and the curriculum learning (Algortihm 1) on the training performance. Sensitivity analysis is a technique that explores the effect of varying hyperparameters to understand their influence on the model's output. Figure 10a shows that increasing the number of steps ahead improves the performance. Increasing the number of steps implies higher memory requirements and longer training times. Because of the best performances and GPU availability, we selected 8 steps ahead in all experiments. However, when performing bigger hyper-parameter searches or when limited by hardware, choosing fewer steps ahead can result in an acceptable performance. Similar considerations can also be done for the CNN model.

Figure 10b shows that increasing the interval of curriculum steps linearly reduces the training times, while also improving the performance. The decrease in performance associated to bigger values is probably caused by the number of total training epochs, i.e., 150, which are insufficient to cover the whole prediction horizon $H$. Increasing the total number of epochs should increase both the performance and the training time but we avoided this analysis and chose an interval of 15 epochs for the curriculum learning strategy, as a trade-off between performance and training times. Moreover, models with curriculum steps between 0 and 15 suffered from spurious instabilities during training, that were compensated with early stopping, while models with more curriculum steps were generally more stable. This is due to sudden variations in the loss function that limit a smoother learning process.

## 6 Concluding remarks

We proposed a deep learning model for rapid flood modelling, called SWE-GNN, inspired by shallow water equations (SWE) and graph neural networks (GNN). The model takes the same inputs as a numerical model, i.e., the spatial discretization of the domain, elevation, slopes, and initial value of the hydraulic variables, and predicts their evolution in time in an auto-regressive manner. The results show that the SWE-GNN can correctly predict the evolution of water depth and discharges with mean average errors in time of $0.04\ m$ and $0.004\ m^2/s$, respectively. It also generalizes well to previously unseen topographies with varying breach locations, bigger domains, and longer time horizons. SWE-GNN is up to two orders of magnitude times faster than the underlying numerical model. Moreover, the proposed model achieved consistently better performances with respect to other deep learning models, in terms of water depth and unit discharge errors as well as CSI.

In line with the hypothesis, GNNs proved to be a valuable tool for spatio-temporal surrogate modelling of floods. The analogy with finite volume methods is relevant for three motivations. First, it improves the deep learning model's interpretability, as the weights in the graph propagation rule can be interpreted as an approximate Riemann solver and multiple GNN layers can be seen as intermediate steps of a multi-step method such as Runge-Kutta. Second, the analogy also provides an existing framework to include conservation laws in the model and links two fields that can benefit from each other advances. For example, multiple spatial and temporal resolutions could be jointly used, in place of a fixed one, similarly to Liu et al. (2022). Third, the methodology is applicable for any flood modelling application where the SWE hold, such as storm surges and river floods. The same reasoning can also be applied to other types of partial differential equations where finite volume methods are commonly used, such as in computational fluid dynamics.

The current analysis was carried out under a constant breach inflow as a boundary condition. Further research should extend the analysis to time-varying boundary conditions to better represent complex real-world scenarios. One solution is to employ ghost cells, typical of numerical models (LeVeque et al., 2002), for the domain boundaries, assigning known values in time. It must be noted that our model cannot yet completely replace numerical models, as it requires the first time step of the flood evolution as input. This challenge could be addressed by directly including boundary conditions in the model's inputs. Contrarily to physically-based numerical methods, the proposed model does not strictly enforce conservation laws, such as mass balance. Future work could address this limitation by adding conservation equations in the training loss function, as is commonly done with physics-informed neural networks. Finally, while we empirically showed that the proposed model along with the multi-step-ahead loss can sufficiently overcome numerical stability conditions, we provide no theoretical guarantee that stability can be enforced for an indefinite amount of time steps.

Future research should investigate the new modelling approach in flood risk assessment and emergency preparation. This implies creating ensembles of flood simulations to reflect uncertainties, flood warning and prediction of extreme events, and exploring adaptive modelling during floods, by incorporating real-time observations. The model should also be validated in real case studies featuring linear elements such as secondary dikes and roads, typical of polder areas. Further work could also address breach uncertainty in terms of timing, size, growth, and amount of breaches. Moreover, future works should aim at improving the model's Pareto front. For improving the speed-up, one promising research direction would be to employ multi-

scale methods that allow to reduce the number of message passing operations, while still maintaining the same interaction range (e.g., Fortunato et al., 2022; Lino et al., 2022). On the other hand, better enforcing physics and advances in GNNs with

spatio-temporal models (e.g., Sabbaqi and Isufi, 2022) or generalizations to higher-order interactions (e.g., Yang et al., 2022) may further benefit the accuracy of the model. Overall, the SWE-GNN marks a valuable step towards the integration of deep learning for practical applications.

## Appendix A: Architecture details

In this section, we further detail the different inputs and outputs, the hyperparameters, and the models' architectures used in

Section 5.1.

### A1   Inputs, outputs, and hyperparameters

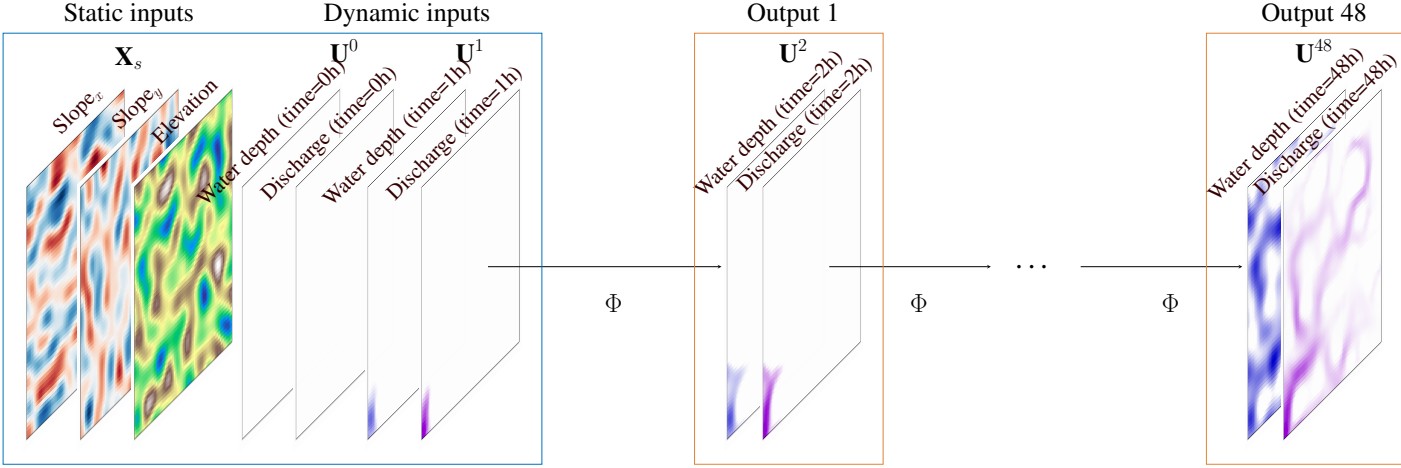

**Figure A1.** Detailed inputs and outputs used in the paper, considering a regular mesh, $p = 1$ previous time steps and a time resolution $\Delta t = 1h$. The initial inputs are dry bed conditions, i.e., $\mathbf{U}^{t=0h}$, and the first time step of the simulation, i.e., $\mathbf{U}^{t=1h}$, given by the numerical model.

    Fig. A1 shows the inputs employed by all models in Section 5.1. The static inputs $\mathbf{X}_s$ are given by the slopes in the $x$ and $y$ directions, and the elevation, while the initial dynamic inputs $\mathbf{X}_d = (\mathbf{U}^0, \mathbf{U}^1)$ are given by water depth and discharge at times $t = 0h$, i.e., the empty domain, and $t = 1h$.

Table A1 shows the hyperparameters employed for each model. Some hyperparameters are common to all models, such as learning rate, number of maximum training steps ahead, and optimizer, while other change depending on the model, such as embedding dimensions and number or layers.

**Table A1.** Summary of the hyperparameters and related values' ranges employed for the different deep learning models. The **bold** values indicate the best configuration in terms of validation loss.

| DL model | Hyperparameter name | Values' range (**best**) |
| --- | --- | --- |
| All models | Initial learning rate | 0.005 |
| | Input previous time steps ($p$) | 1 |
| | Temporal resolution ($\Delta t$) | $1h$ |
| | Maximum training steps ahead ($H$) | 8 |
| | Optimizer | Adam |
| GNN models | Embedding dimension ($G$) | 8,16,32,**64** |
| | Number of GNN layers ($L$) | 1,2,3,4,5,6,7,**8**,9 |
| | Batch size | 8 |
| CNN | First embedding dimension | 16,32,**64**, 128 |
| | Number of encoding blocks | 1,2,**3**,4 |
| | Activation function | ReLU, **PReLU**, no activation |
| | Batch size | 64 |

## A2  GNN benchmarks

We compared the proposed model against two benchmark GNNs that employ different propagation rules. Since those models cannot independently process static and dynamic attributes, contrarily to the SWE-GNN, we stacked the node inputs into a single node feature matrix $\mathbf{X} = (\mathbf{X}_d, \mathbf{X}_s)$, which passes through an encoder MLP and then to the GNN.

**Graph Convolutional Neural Network (GCN)** employs the normalized Laplacian connectivity matrix to define the edge weights $\mathbf{s}_{ij}$. The layer propagation rule reads as:

$$s_{ij} = \left(\mathbf{I} - \mathbf{D}^{-1/2}\mathbf{A}\mathbf{D}^{-1/2}\right)_{ij}, \tag{A1}$$

$$\mathbf{h}_i^{(\ell+1)} = \sum_{j \in \mathcal{N}_i} s_{ij}\mathbf{W}^{(\ell)}\mathbf{h}_j^{(\ell)}, \tag{A2}$$

where $\mathbf{I}$ is the identity matrix, $\mathbf{A}$ is the adjacency matrix, which has non-zero entries in correspondence of edges, and $\mathbf{D}$ is the diagonal matrix.

**Graph Attention Network (GAT)** employs an attention-based mechanism to define the edge weights $\mathbf{s}_{ij}$ based on their importance in relation to the target node. The layer propagation rule reads as:

$$\mathbf{s}_{ij} = \frac{exp(LeakyReLU(\mathbf{a}^T[\mathbf{W}^{(\ell)}\mathbf{h}_i^{(\ell)}||\mathbf{W}^{(\ell)}\mathbf{h}_k^{(\ell)}]))}{\sum_{k \in N_i} exp(LeakyReLU(\mathbf{a}^T[\mathbf{W}^{(\ell)}\mathbf{h}_i^{(\ell)}||\mathbf{W}^{(\ell)}\mathbf{h}_k^{(\ell)}]))}, \tag{A3}$$

$$\mathbf{h}_i^{(\ell+1)} = \sum_{j \in \mathcal{N}_i} \mathbf{s}_{ij}\mathbf{W}^{(\ell)}\mathbf{h}_j^{(\ell)}, \tag{A4}$$

where $\mathbf{a} \in \mathbb{R}^{2G}$ is a weight vector, $\mathbf{s}_{ij}$ are the attention coefficients, and $||$ denotes concatenation.

## A3  CNN

The encoder-decoder convolutional neural network is an architecture composed of two parts (Fig. A2). The encoder extracts high-level features from the input images, while reducing theirs extent, via a series of convolutional and pooling layers, while the decoder extracts the output image from the compressed signal, again via a series of convolutional layers and pooling layers. The U-NET version of the architecture also features residual connections between images with the same dimensions, i.e., the output of an encoder block is summed to the inputs of the decoder block with the same dimensions, as shown in Fig. A2. The equation for a single 2D convolutional layer is defined as:

$$\mathbf{Y}_k = \sigma(\mathbf{W}_k * \mathbf{X}), \tag{A5}$$

where $\mathbf{Y}_k$ is the output feature map for the $k$-th filter, $\mathbf{X}$ is the input image, $\mathbf{W}_k$ is the weight matrix for the $k$-th filter, $*$ denotes the 2D convolution operation, and $\sigma$ is an activation function.

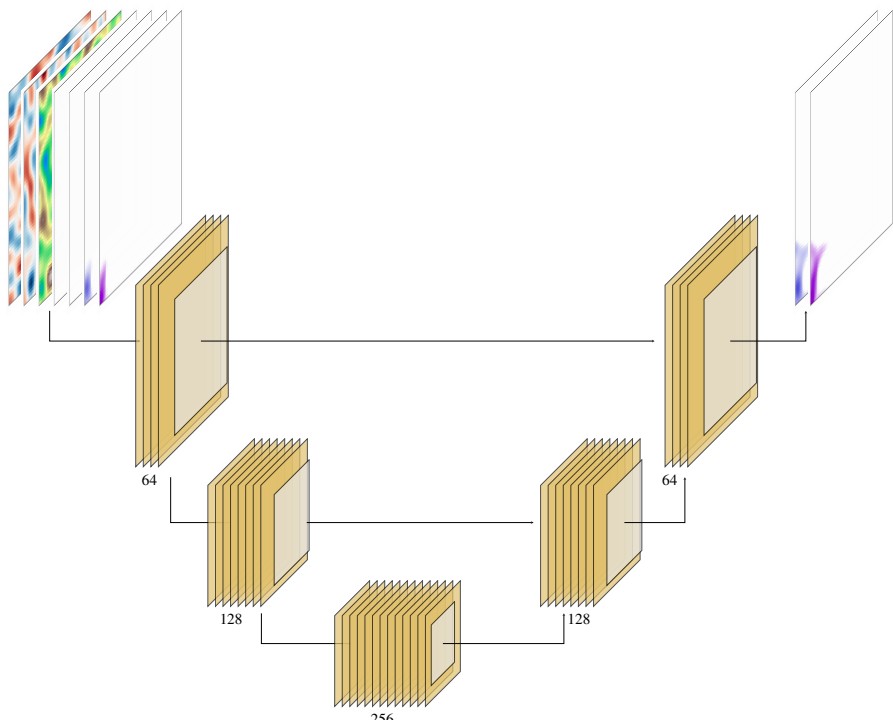

**Figure A2.** U-NET based CNN architecture employed in the experiments, with first embedding dimension of 64 and three encoding blocks. Each block is composed of one convolutional layer, followed by a batch normalization layer, a *PReLU* activation function, another convolutional layer, and finally a pooling layer. All blocks with the same dimensions are connected by residual connections, indicated by the horizontal lines.

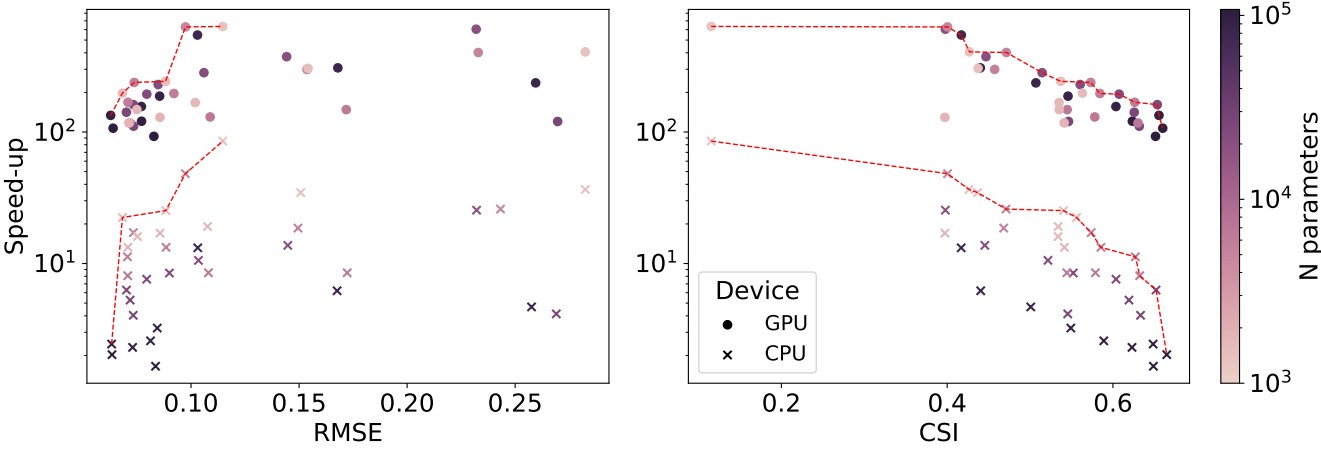

**Figure B1.** Pareto fronts on test dataset 3 (red-dotted lines) in terms of speed-ups, RMSE, and CSI for varying number of parameters for a temporal resolution Δt=1h.

## Appendix B: Pareto front for dataset 3

We employed the models trained with different combinations of number of GNN layers and embedding size (Section 5.3) on test dataset 3. Figure B1 shows that the models performs better in terms of speed with respect to the smaller areas, achieving similar CPU speedups and GPU speedups around two times higher than those in datasets 1 and 2.

*Code and data availability.* The employed dataset can be found at https://dx.doi.org/10.5281/zenodo.7764418. The code repository is available at https://github.com/RBTV1/SWE-GNN-paper-repository-.

*Video supplement.* The simulations on the test datasets 1, 2, and 3, run with the presented model, can be found at https://dx.doi.org/10.5281/zenodo.7652663.

*Author contributions.* **Roberto Bentivoglio**: Conceptualization, Methodology, Software, Validation, Data curation, Writing- Original draft preparation, Visualization, Writing - Review & Editing. **Elvin Isufi**: Supervision, Methodology, Writing - Review & Editing, Funding acquisition. **Sebastiaan Nicolas Jonkman**: Supervision, Writing - Review & Editing. **Riccardo Taormina**: Conceptualization, Supervision, Writing - Review & Editing, Funding acquisition, Project administration.

*Competing interests.* No competing interests are present.

*Acknowledgements.* This work is supported by the TU Delft AI Initiative program. We thank Ron Bruijns for providing the dataset to carry out the preliminary experiments. We thank Deltares for providing the license for Delft3D to run the numerical simulations.

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
