# Peer review of "Rapid Spatio-Temporal Flood Modelling via Hydraulics-Based Graph Neural Networks"

_EGUsphere, 2023_

## Author Comment (AC1)

We thank the Reviewer for the helpful comments and suggestions. We hereby address them individually. In this document we indicate the *Reviewer's comments in italic dark grey*, while text that was changed in the paper in blue.

*This paper introduces a novel machine learning algorithm for the simulation of 2D surface flooding. The algorithm takes inspiration from scientific computing techniques to develop the machine learning architecture, to arrive at a setup that provides a time-dynamic and spatially distributed simulation, and a speedup in the order of factor 100 compared to a numerical simulation model. To my knowledge, this is the first application of this kind of methodology for flood problems. A number of questions therefore remain unsolved, and the considered case examples do not yet reflect all complexities that we encounter in the real world. On the other hand, this work is likely to inspire a variety of derivative works in hydrology, where the potential impact of these techniques so far is not really recognized. Because this work is a frontrunner, I have a number of suggestions that are mainly aimed at making the work accessible to a broader audience. I think these can be addressed in a minor revision, and I recommend the paper for publication, subject to these modifications.*

We thank the Reviewer for the comments and for sharing our view on the potential impact of this paper. Regarding the accessibility to a broader audience, we will also release a tutorial notebook on how to use the model alongside with the code repository that will be shared upon paper publication.

*Main comments:*

*Comment 1: Limitations - and important feature of numerical methods is that they (mostly) preserve e.g. mass and momentum. As far as I can see, this is not the case for the algorithm proposed here, and it is not straightforward to see how this can be implemented in the encoder/decoder architecture. This should be clearly mentioned as a limitation.*

Answer 1: We thank the Reviewer for the comment. We included the following additional sentences in the discussion to emphasize this limitation and how it could be addressed.

Lines 415-417: "Contrarily to physically-based numerical methods, the proposed model does not strictly enforce conservation laws, such as mass balance. Future work could address this limitation by adding conservation equations in the training loss function, as is commonly done with physics-informed neural networks."

*Comment 2: A major challenge with this type of model is actually implementation. For example, efficient data pipelines for custom graph operators are not a straightforward problem. I would strongly suggest a section or appendix summarizing the main computational challenges and how you suggest to address them.*

Answer 2: We understand the concern with graph operators not being commonly used in the water field, however there are now several libraries, such as pytorch geometric, which offer a great variety of implementation strategies which are relatively easy to follow using the documentation. Moreover, GNN-based solutions have been implemented in billion-scale graphs (D. Zheng et al., 2020) and there are also companies working on specific hardware for graph-based NNs (e.g., Graphcore). So, overall, we believe that this might not be so big of an issue.

We highlighted the implementation issue in lines 276-277 as follows:

"Pytorch Geometric is a library that allows easy implementation of graph neural networks, also for billion-scale graphs (D. Zheng et al., 2020)."

D. Zheng *et al.*, "DistDGL: Distributed Graph Neural Network Training for Billion-Scale Graphs," *2020 IEEE/ACM 10th Workshop on Irregular Applications: Architectures and Algorithms (IA3)*, GA, USA, 2020, pp. 36-44, doi: 10.1109/IA351965.2020.00011.

https://www.graphcore.ai/posts/accelerating-pyg-on-ipus-unleash-the-power-of-graph-neural-networks?utm_content=243840010&utm_medium=social&utm_source=linkedin&hss_channel=lcp-10812092

*Comment 3: Along the same lines, I appreciate that the authors have tried to keep the Methodology description generic. However, this also makes it very hard to read in some places. I think you could greatly help the readers with an Appendix that includes a detailed variant of Figure 2, where you include equations 5 to 10, and where hyperparameters (G, p, etc.) reflect that actual values used in your implementation.*

Answer 3: As suggested by the Reviewer, we included in Appendix A a more detailed description of the inputs and outputs used by the model, in the specific case of a temporal resolution of 1 hour and with only one previous time step as input. To better explain this, we added the following figure:

[Figure]

**Figure A1.** Detailed inputs and outputs used in the paper, considering a regular mesh, $p = 1$ previous time steps and a time resolution $\Delta t = 1h$. The initial inputs are dry bed conditions, i.e., $\mathbf{U}^{t=0h}$, and the first time step of the simulation, i.e., $\mathbf{U}^{t=1h}$, given by the numerical model.

We describe Figure A1 in the appendix (lines 431-433) as follows:

"Fig. A1 shows the inputs employed by all models in Section 5.1. The static inputs Xs are given by the slopes in the x and y directions, and the elevation, while the initial dynamic inputs Xd = (U0, U1) are given by water depth and discharge at times t = 0h, i.e., the empty domain, and t = 1h."

Following also the minor comments (comment 7), we added the equation numbers directly in Figure 2, as later descripted.

*Comment 4: Regarding the hyperparameters, I strongly miss a table that summarizes*

*the final set of hyperparameters. These are now spread out through the paper. In addition, the complexity of the different MLPs used throughout the methodology is currently entirely unclear.*

Answer 4: We thank the Reviewer for the comment. We added a table summarizing the hyperparameters and their values' range in Appendix A. Lines 435-437 read as follows:

"Table A1 shows the hyperparameters employed for each model. Some hyperparameters are common to all models, such as learning rate, number of maximum training steps ahead, and optimizer, while other change depending on the model, such as embedding dimensions and number or layers."

**Table A1.** Summary of the hyperparameters and related values' ranges employed for the different deep learning models. The **bold** values indicate the best configuration in terms of validation loss.

| DL model | Hyperparameter name | Values' range (**best**) |
|---|---|---|
| All models | Initial learning rate | 0.005 |
| | Input previous time steps ($p$) | 1 |
| | Temporal resolution ($\Delta t$) | $1h$ |
| | Maximum training steps ahead ($H$) | 8 |
| | Optimizer | Adam |
| GNN models | Embedding dimension ($G$) | 8,16,32,**64** |
| | Number of GNN layers ($L$) | 1,2,3,4,5,6,7,**8**,9 |
| | Batch size | 8 |
| CNN | First embedding dimension | 16,32,**64**, 128 |
| | Number of encoding blocks | 1,2,**3**,4 |
| | Activation function | ReLU, **PReLU**, no activation |
| | Batch size | 64 |

Regarding the MLP characterisation, we added the following sentences in Section 3.1.

Lines 153-154: "All MLPs have two layers, with hidden dimension G, followed by a PReLU activation."

Lines 168-169: "where $\psi(\cdot)$ : R5G → RG is an MLP with two layers, with hidden dimension 2G, followed by a PReLU activation function, $\odot$ is the Hadamard (element-wise) product, and $W(\ell) \in$ RG×G are parameter matrices."

Lines 192: "The MLP $\varphi(\cdot)$ has two layers, with hidden dimension G, followed by a PReLU activation."

*Comment 5: The benchmark models are introduced in a few lines of text, and not easily understood. Also here, I suggest including an appendix that details these.*

Answer 5: As requested also in point 3, we detailed all model architectures and benchmark models in Appendix A, including explanatory figures and equations.

Appendix A reads as follows:

"In this section, we further detail the different inputs and outputs, the hyperparameters, and the models' architectures used in Section 5.1.

**A1 Inputs, outputs, and hyperparameters**

See answers 3 and 4

**A2 GNN benchmarks**

We compared the proposed model against two benchmark GNNs that employ different propagation rules. Since those models cannot independently process static and dynamic attributes, contrarily to the SWE-GNN, we stacked the node inputs into a single node feature matrix X = (Xd, Xs), which passes through an encoder MLP and then to the GNN.

Graph Convolutional Neural Network (GCN) employs the normalized Laplacian connectivity matrix to define the edge weights sij. The layer propagation rule reads as:

$$s_{ij} = \left( \mathbf{I} - \mathbf{D}^{-1/2}\mathbf{A}\mathbf{D}^{-1/2} \right)_{ij}, \tag{A1}$$

$$\mathbf{h}_i^{(\ell+1)} = \sum_{j \in \mathcal{N}_i} s_{ij} \mathbf{W}^{(\ell)} \mathbf{h}_j^{(\ell)}, \tag{A2}$$

where I is the identity matrix, A is the adjacency matrix, which has non-zero entries in correspondence of edges, and D is the diagonal matrix.

Graph Attention Network (GAT) employs an attention-based mechanism to define the edge weights $s_{ij}$ based on their importance in relation to the target node. The layer propagation rule reads as:

$$\mathbf{s}_{ij} = \frac{exp(LeakyReLU(\mathbf{a}^T[\mathbf{W}^{(\ell)}\mathbf{h}_i^{(\ell)}||\mathbf{W}^{(\ell)}\mathbf{h}_k^{(\ell)}))}{\sum_{k \in N_i} exp(LeakyReLU(\mathbf{a}^T[\mathbf{W}^{(\ell)}\mathbf{h}_i^{(\ell)}||\mathbf{W}^{(\ell)}\mathbf{h}_k^{(\ell)}]))}, \tag{A3}$$

$$\mathbf{h}_i^{(\ell+1)} = \sum_{j \in \mathcal{N}_i} \mathbf{s}_{ij} \mathbf{W}^{(\ell)} \mathbf{h}_j^{(\ell)}, \tag{A4}$$

where $\mathbf{a} \in R^{2G}$, is a weight vector, $s_{ij}$ are the attention coefficients, and || denotes concatenation.

**A3 CNN**

[Figure]

**Figure A2.** U-NET based CNN architecture employed in the experiments, with first embedding dimension of 64 and three encoding blocks. Each block is composed of one convolutional layer, followed by a batch normalization layer, a *PReLU* activation function, another convolutional layer, and finally a pooling layer. All blocks with the same dimensions are connected by residual connections, indicated by the horizontal lines.

The encoder-decoder convolutional neural network is an architecture composed of two parts (Fig. A2). The encoder extracts high-level features from the input images, while reducing theirs extent, via a series of convolutional and pooling layers, while the decoder extracts the output image from the compressed signal, again via a series of convolutional layers and pooling layers. The U-NET version of the architecture also features residual connections between images with the same dimensions, i.e., the output of an encoder block is summed to the inputs of the decoder block with the same dimensions, as shown in Fig. A2.

The equation for a single 2D convolutional layer is defined as:

$$\mathbf{Y}_k = \sigma(\mathbf{W}_k * \mathbf{X}), \tag{A5}$$

where Yk is the output feature map for the k-th filter, X is the input image, Wk is the weight matrix for the k-th filter, $*$ denotes the 2D convolution operation, and σ is an activation function."

*Detailed comments:*

*Comment 6:line 106: a is already used as symbol for area, use v for velocity?*

Answer 6: We corrected the symbol to *v*, as suggested by the Reviewer, to avoid confusion with the symbol for area.

*Comment 7: Figure 2:*

*-top part of the figure*

*-mention what are the static inputs and the outputs in your work in the figure*

*-include a recursive arrow from output 1 back to dynamic inputs, to make it clear that the model prediction is recycled*

*-bottom part of the figure*

*-include the following captions above MLP, GNN and MLP: "process individual nodes", "process neighborhood of each node", "process individual nodes"*

*-consider referencing the equation numbers inside the figure, to make it clear which illustration relates to what step*

*-include j in one of the orange circles*

*-edge feature encodings should receive and output arrow from the MLPs as well. As a whole, maybe this figure would be easier to understand if you distinguish h_si, h_di (why not h_dy?) and eps_ij separately (three sets of arrows)*

*-caption*

*-h_si and h_di are not clear from the caption (and explained somewhere much later in the main text)*

Answer 7: We greatly thank the Reviewer for the suggestions. We modified figure 2 as follows:

[Figure]

We also modified the caption to better explain what h_si and h_di are. The caption of Figure 2 now reads as:

"… .The node inputs $x_{si}$ and $u^{t-p:t}_i$ represent static attributes, such as elevation and slopes, and dynamic attributes, representing hydraulic variables, while the edge inputs $\varepsilon_{ij}$ represent the mesh's geometry. The inputs are encoded into higher-dimensional embeddings $h_{si}$, $h^0_{di}$ (yellow nodes), and $\varepsilon'_{ij}$ via three separate multi-layer perceptrons, shared across nodes or edges. The embeddings, whose purpose is to increase the inputs' expressivity, are used as input for the L GNN layers. …"

Regarding the terminology used for h_di, we avoided using h_dy as the second subscript term, i.e., *i*, refers to node i, while d stands for dynamic. We clarified this by modifying lines 156-158 as follows:

"The ith rows of the node matrices Hs and Hd represent the encoded feature vectors associated to node i, i.e., hsi and hdi, and the kth rows of the edge matrices E' represents the encoded feature vector associated to edge k."

*Comment 8: line 138: explain that input features and hyperparameters will be explained in section 3.2*

Answer 8: We modified lines 137-138 as suggested by the Reviewer as follows:

"In the following, we detail the proposed model (Section 3.1) and its inputs and outputs (Section 3.2). Finally, we discuss the training strategy (Section 3.3)."

*Comment 9: line 145: Ut−p:t are the dynamic node features --> Ut−p:t are the dynamic node features (hydraulic variables) for time steps t-p to t*

Answer 9: We corrected line 145 following the Reviewer's suggestion.

Line 145: "… Ut−p:t are the dynamic node features, i.e., the hydraulic variables for time steps t-p to t, …"

*Comment 10: line 148: define I_epsilon*

Answer 10: We have defined I_epsilon in line 151, just below it. Thanks for pointing this out.

*Comment 11: line 154: explain that G is a hyperparameter*

Answer 11: We thank the Reviewer for the comment. We explicitly mentioned that G is a hyperparameter in line 155-156 as:

"The encoders expand the dimensionality of the inputs to allow for higher expressivity, with the hyperparameter G being the dimension of the node embeddings."

*Comment 12: Equation 7: At this point it would be really good to know how terrain differences come into play, and you don't explain it until Section 3.2. I think a short explanation is needed here, because it actually hinders the understanding of the methodology*

Answer 12: We thank the Reviewer for the suggestion. Terrain differences come into play in the physical model but not explicitly in this deep learning model. Along with the other static variables and the hydraulic variables, however, they create an estimate of the contribution of the source terms used in the numerical models. This is linked with the interpretation of the MLP in Eq. 7 as a Riemann solver, according to which the output of the MLP is an approximation of the Jacobian of the fluxes. We added a short explanation on the role of function ψ in equation 7 in lines 171-172 as:

"The function ψ(·), instead, incorporates both static and dynamic inputs and provides an estimate of the source terms acting on the nodes."

*Comment 13: line 184: is the activation function only applied on the final graph layer?*

Answer 13: Yes, the nonlinearity is applied only in the final graph layer because preliminary experiments showed comparable or worse performances when using multiple activations.

*Comment 14: line 191: Do the neural networks in the graph layers include bias terms? You refer to sparsity in multiple places in the paper, but you never explain it and why it is relevant (a large area of the image is 0 and does not need to be processed?)*

Answer 14: Yes, the MLPs inside the graph layers (cfr. Eq. 7) use bias terms as most input terms in the function ψ already have non-zero values and thus excluding these

terms would have no marked positive influence. This is not the case for dynamic node features since adding a bias term would result in elements without water, and thus with only zero values, to be non-zero.

To clarify, we modified lines 194-196 as follows: "Both the MLPs in the dynamic encoder and the decoder do not have the bias terms as this would result in adding non-zero values in correspondence of dry areas that would cause water to originate from any node."

We also modified lines 204-205, that mentioned the term sparsity, as:

"The reason why we include $w_i^t$ in the static attributes instead of the dynamic ones is that this features can be non-zero also without water, due to the elevation term, and would thus result in the same issue mentioned for the dynamic encoder and decoder."

*Comment 15: Eq. 12: it is confusing that u is not the same vector as when introducing the SWE. I have no good suggestion for improving this though.*

Answer 15: We thank the Reviewer for the suggestion. We clarified the difference between the u used in our deep learning model and that defined in Eq. 2. Lines 211-214 now read as follows:

"Contrarily to the definition of the hydraulic variables as in Eq. (2), we selected the modulus of the unit discharge |q| as a metric of flood intensity in place of its x and y components to avoid mixing scalar and vector components and because, for practical implications, such as damage estimation, the flow direction is less relevant than its absolute value (e.g., Kreibich et al., 2009)."

*Comment 16: line 212: define unit normal vector (probably already needed in the context of Fig. 2)*

Answer 16: We defined the unit normal vector in line 102 where it appears for the first time.

*Comment 17: Section 3.3: Consider merging this with 4.2. Searching for the actual parameters used in training is yet another unnecessary complication*

Answer 17: We understand the Reviewer's concern. However, we decided to keep the two sections separate as Section 3.3 describes a generic methodology that can be applied as well to different applications, while 4.2 describes the specific implementations used in the papers' setting.

*Algorithm 1: Nice!*

Thanks!

*Comment 18: line 266: I don't understand why H is now fixed (previously, you introduced the curriculum). Is this the maximum of H considered?*

Answer 18: We thank the Reviewer for comment. Indeed with this H we intended the maximum value that can be reached during training. Hence, we modified lines 271-272 as:

"We used a maximum prediction horizon H=8 steps ahead during training as a trade-off between model stability and training time."

*Comment 19: Section 4.3: I think it would be interesting to see some illustrations of how e.g. MAE evolves over simulation time. This would help us understand better if the method is stable*

Answer 19: We thank the Reviewer for the suggestion. We included a new figure (Figure 7) to represent how MAE, RMSE, and CSI evolve in time over the whole testing dataset. We added lines 359-363 as:

"We also observe the average performance of the different metrics over time, for the whole test dataset 1, in Figure 7. The CSI is consistently high throughout the whole simulation, indicating that the model correctly predicts where water is located in space and time. On the other hand, both MAE and RMSE increase over time. This is partially due to the evaluation of both metrics via a spatial average, which implies that in the first time steps, where the domain is mostly dry, the error will naturally be lower. Nonetheless, the errors increase linearly or sub-linearly, implying that they are not prone to explode exponentially."

[Figure]

**Figure 7.** Temporal evolution of CSI scores, MAE, and RMSE for test dataset 1. The confidence bands refer to one standard deviation from the mean.

*Comment 20: line 298: I'm not sure what propagation rule you refer to. As mentioned above, these model variations need to be documented better.*

Answer 20: Since we added an appendix section with more details on the methods, we added the following sentences for both GAT and GCN in lines 305 and 307:

"For more details see Appendix A."

*Comment 21: Figure 5: Show units also on the legends of the difference plots*

Answer 21: We thank the Reviewer for the comment. We modified Figure 5 accordingly as:

[Figure]

*Comment 22: Figure 7+8: From these figures, speedup in the order of factor 100 seems more realistic to me (not 600 as mentioned somewhere in the paper)*

Answer 22: We agree with the Reviewer that for dataset 1 the order of 100 is a more realistic value than 600. The speedup range of 100-600 comes from running the same Pareto-front in Figure 8 on dataset 3. Here the model better exploits the bigger domain size and produces a higher range of speedup values. We included this figure, along with further details on it in Appendix B, which reads as follows:

"Appendix B: Pareto front for dataset 3

[Figure]

**Figure B1.** Pareto fronts on test dataset 3 (red-dotted lines) in terms of speed-ups, RMSE, and CSI for varying number of parameters for a temporal resolution Δt=1h.

We employed the models trained with different combinations of number of GNN layers and embedding size (Section 5.3) on test dataset 3. Figure B1 shows that the models performs better in terms of speed with respect to the smaller areas, achieving similar

CPU speedups and GPU speedups around two times higher than those in datasets 1 and 2."

---

## Author Comment (AC2)

We thank the Reviewer for his careful review and appreciation of our work. We value his insightful comments and suggestions, which we hereby address individually. In this document we indicate the *Reviewer's comments in italic dark grey*, while text that was changed in the paper in blue.

*This paper presents a ML-based approach for two-dimensional hydrodynamic modeling of floodings. In my eyes, the design takes inspiration from Geometric Deep Learning to wisely choose inductive biases so that the presented approach is aligned with knowledge from physical-principles (e.g., interactions follow along a gradient) and numerical modeling (e.g., the depth of the network compensates for time resolution). The result is a graph-based approach that is two magnitudes faster than a comparable numerical model and its performance is as good or better as other baselines — at least in modelland. The ideas are novel, the evaluation is good, and the exposition is clear. I only have small nitpicks and hope that the paper will be published as soon.*

We are happy to see the Reviewer appreciates the design inspirations and we thank him for the kind words.

*General Comments*
*1) Ablation Study. I have two problems with the ablation. First: I think that the major part of the Copernicus audience will (sadly) not be familiar with the term and I would thus propose to introduce what an "ablation study" is and what you do there as part of your experimental setup (e.g., after Section 4.3). Second: I will emphasize that I very, very much appreciate that the study did ablations. Ablations have become one of the primary tools in ML research, and I believe that many of the current Deep Learning applications are sourly missing them. Still, I want to attest that the presented ablations are rather unusual. An ablation usually refers to an experiment that removes part of the network. Here, on the other hand, the authors did ablate the loss and the algorithm. I am not sure if it is necessary to deal with this minor idiosyncrasy, but I personally would not call this an ablation at all, but rather an exploration of the importance of hyper-parameter choice (I do admit that both are very similar in intent). On the other hand, if the authors like their ablation-framing, I would like to suggest to also include an ablation that checks what happens if one reduces the model itself (e.g., removing the activations in equation 8 PRELU -> RELU -> no activation; or reducing the inductive bias by not using the h-h part in equation 7).*

1) We thank the Reviewer for noticing this possible unfamiliarity with the term. According to both parts of the comment, we changed its name into "sensitivity analysis" and we added an explanation of its purpose in lines 378-380 as:

"Finally, we performed a sensitivity analysis on the role of the multi-step-ahead function (cfr. eq. (14)) and the curriculum learning (Algortihm 1) on the training performance. Sensitivity analysis is a technique that explores the effect of varying hyper-parameters to understand their influence on the model's output."

Regarding the suggested ablations, we already included the effect of removing the h-h part of equation 7 in section 5.1 (the model called SWE-GNN_ng). On the other hand, we believe that changing the nonlinearity would be somewhat tangential to the core message our paper, particularly after clarifying what we meant by ablation. That said, we will gladly consider the effect of different (or no) activation functions in our future work, and we thank again the reviewer for his insights.

*2) Future work on speeding the model further up. I think a discussion on further speed ups of the modeling pipeline would be good. Is a two-magnitude speed up already so much that its not worth pursuing even faster models? Can we still expect speedups? How important is the tradeoff between the temporal resolution and the speed mentioned in L. 260? Etc. Here is, for example, a direction I spontaneously see (not saying that this should be included; just for providing inspiration): In theory one can directly output multiple (all) needed timesteps from the network. This would likely speed the process up considerably, since no recurrence over time (as shown, e.g., in Figure 3) would be required anymore. The price one pays then is that this "concurrent output" approach can only model the inundation for a finite time horizon (as a matter of fact, it will always model exactly to that horizon no matter what). One can also think of "in-between solutions' ', in which the model ingests and outputs chunks of time. This is not so easy with graphs, and does break the nice alignment with the physical conception of the problem.*

2) We thank the Reviewer for the valuable feedback. Indeed, there are still many possibilities to speed-up the current model. While techniques like the one suggested here have already been applied (e.g., Brandstetter et al. 2022), we believe that it would not entirely solve the issue, as predicting more steps at the same time would still imply using more GNN layers, which are the true bottleneck of the model. Accordingly, assuming that the key factor is the number of GNN layers (for a given prediction horizon), one promising research direction would be to employ multi-scale methods that allow to reduce the number of message passing operations, while still maintaining the same interaction range. Since there is always a trade-off between speed and accuracy, we decided to further expand the suggestion by discussing speed-up in combination with finding a better Pareto front that would also results in better trade-offs.

As such, we included lines 428-431 as:

"Moreover, future works should aim at improving the model's Pareto front. For improving the speed-up, one promising research direction would be to employ multi-scale methods that allow to reduce the number of message passing operations, while still maintaining the same interaction range (e.g., Fortunato et al., 2022; Lino et al., 2022). On the other hand, better enforcing physics and advances in GNNs with spatio-temporal models (e.g., Sabbaqi and Isufi, 2022) or generalizations to higher-order interactions (e.g., Yang et al., 2022) may further benefit the accuracy of the model."

Brandstetter, J., Worrall, D. and Welling, M., 2022. Message passing neural PDE solvers. arXiv preprint arXiv:2202.03376.

Fortunato, M., Pfaff, T., Wirnsberger, P., Pritzel, A. and Battaglia, P., 2022. Multiscale meshgraphnets. arXiv preprint arXiv:2210.00612.

Lino, M., Fotiadis, S., Bharath, A.A. and Cantwell, C.D., 2022. Multi-scale rotation-equivariant graph neural networks for unsteady Eulerian fluid dynamics. Physics of Fluids, 34(8), p.087110.

*Specific Comments*
*3) L. 16. I think I know what the authors want to say here, but in general I think that it is not clear what "bigger domains" and "longer periods" of time mean here. Bigger and longer than what?*

3) Following the comment, we modified lines 16-17 as:
"Moreover, it generalizes well to unseen breach locations, bigger domains, and over longer periods of time, compared to those of the training set, outperforming other deep learning models"

*4) L. 71. Maybe it would be good to say "two orders of magnitude" instead of "up to 600 times speed-ups" to align the contribution with the abstract.*

4) We modified line 71 as suggested by the Reviewer as:
"We show that the proposed model can surrogate numerical solvers for spatio-temporal flood modelling in unseen topographies and unseen breach locations, with two orders of magnitude speed-ups"

*5) L.121. I would suggest to remove the phrasing "... well-known 'curse of dimensionality', ...", because (1) it might not be as well known as you think, and perhaps more importantly (2) the term "curse of dimensionality" refers to a plethora of phenomena and thus readers might associate something different with it. Instead, I would propose to write something direct like "For MLPs the number of parameters increases exponentially with ..."*

5) We thank the Reviewer for the relevant observation. As suggested, we modified lines 120-121 as:
"For MLPs, the number of parameters and the computational cost increase exponentially with the dimensions of the input."

*6) L. 126. Bronstein (2021) as a sole reference is probably a bit unfitting here, since LeCun and Bengio (1995) already discussed the importance of shared weights in CNNS (according to ideas lined out by LeCun in 1989). References:*
*- LeCun, Y., & Bengio, Y. (1995). Convolutional networks for images, speech, and time series. The handbook of brain theory and neural networks, 3361(10), 1995.*
*- LeCun, Y. (1989). Generalization and network design strategies. Connectionism in perspective, 19(143-155), 18.*

6) We included LeCun et al. (2015) as a further reference in line 126. We opted for this reference instead of the suggested ones, as it is more recent.

LeCun, Y., Bengio, Y. and Hinton, G., 2015. Deep learning. nature, 521(7553), pp.436-444.

*7) L. 131. The word "avoid" might be misleading here, as they just don't include these specific physical inductive biases. Maybe use "include" instead.*

7) Indeed, we agree with the Reviewer to changing the term "avoid" into "do not include" as it fits better in this context.

*8) L. 167f. "Either" here would suggest that both node-features need to be non-zero. I don't see that. Is this a formulation thing or am I missing something?*

8) In this sentence, with "either of the interfacing node features is non-zero" we mean that at least one of the two needs to be non-zero. To avoid confusions, we changed the term "either" with "at least one".

*9) L. 174. More a thought than a suggestion: For me the W(I) is like an additional MLP layer (that maps to R1) without an activation. It might thus also be interesting to do an ablation on W(I).*

9) We agree with the Reviewer that the weight matrix W could be replaced as well with a more complex function such as an MLP, as proposed as well in other works (e.g., Battaglia et al., 2018). Since preliminary results showed comparable results, we decided avoiding adding this component in favour of the proposed one, as the latter given more interpretability to what each component represents.

Battaglia, P.W., Hamrick, J.B., Bapst, V., Sanchez-Gonzalez, A., Zambaldi, V., Malinowski, M., Tacchetti, A., Raposo, D., Santoro, A., Faulkner, R. and Gulcehre, C., 2018. Relational inductive biases, deep learning, and graph networks. arXiv preprint arXiv:1806.01261.

*10) L. 191. Can you clarify this sentence? I understand that you don't want areas with constant values in it, but in my opinion the sentence is formulated a bit vaguely.*

10) As suggested as well by Reviewer 1, we modified this line as:
"Both the MLPs in the dynamic encoder and the decoder do not have the bias terms as this would result in adding non-zero values in correspondence of dry areas that would cause water to originate from any node."

*11) Algorithm 1. I think you should at least initialize the weighting coefficients and the CurriculumSteps variable since they are used so explicitly (technically the other variables would need to be chosen too, but for them I think it's less crucial).*

11) We thank the Reviewer for the suggestion. We modified Algorithm 1, including the initialization of weights and curriculum steps, as recommended.

**Algorithm 1** Curriculum learning strategy

**Initialize:**
  $H = 1$
  $CurriculumSteps = 15$
  $\gamma_1 = 1$ (Water depth $h$)
  $\gamma_2 = 3$ (Unit discharge $q$)
**for** epoch = 1 to MaxEpochs **do**
  $\hat{\mathbf{U}}^{t+1} = \mathbf{U}^t + \Phi(\mathbf{X}_s, \mathbf{U}^{t-p:t}, \mathcal{E})$
  $\mathcal{L} = \frac{1}{HO} \sum_{\tau=1}^{H} \sum_{o=1}^{O} \gamma_o \|\hat{\mathbf{u}}_o^{t+\tau} - \mathbf{u}_o^{t+\tau}\|_2,$
  Update the parameters
  **if** epoch > CurriculumSteps*H **then**
    $H = H + 1$
  **end if**
**end for**

*12) L. 266. This should refer to the ablation study to show that the choice of H=8 is not arbitrary.*

12) We thank the Reviewer for the suggestion. We included a reference to the sensitivity analysis in lines 266-267 as:

"We used a maximum prediction horizon H=8 steps ahead during training as a trade-off between model stability and training time, as later highlighted in Section 5.4."

*13) L. 267f. The sentence is a bit peculiar and maybe I missed it: I see that discharge is weighted with 3, but what is the weighting factor for the water depth (I assume 30)?*

13) We understand that the considerations on the weighting factor are discussed rapidly. The motivation for weighting the different output components is that there in no normalization performed as pre-processing step and, as such, the values of water depth and unit discharge generally differ in magnitude by a factor of 10. Thus, logically, we would weight the discharge by a factor of 10 to achieve a similar goal as by normalizing them. However, we also consider that for application purposes water depth is more relevant that discharge. Consequently, we opted for a smaller weighting factor whose value of 3 was selected after some preliminary analysis.
Hence, we modified lines 267-268 to clarify this issue as:
"There is no normalization pre-processing step and, thus, the values of water depth and unit discharge differ in magnitude by a factor of 10. Since for application purposes discarge is less relevant than water depth (Kreibich et al., 2009), we weighted the discharge term by a factor of γ2 = 3 (cfr. eq. (14)), while leaving the weight factor for water depths as γ1 = 1."

*14) L. 277. MAE: I guess technically this is the mean of the mean absolute errors per variable, since you calculate the error in u_o (i.e., the Mean of the L1 for each hydraulic variable) and not in u (i.e. the L1 norm over the vector that spans all hydraulic variables). For me your choice makes more sense anyways. More importantly: What is of interest to me here, would be to see how the MAE changes if you include the weighting factors of the loss in the evaluation and get an weighted MAE (so that water depth is more important).*

14) We thank the Reviewer for pointing this out. There is a mistake in how the testing MAE was presented: as highlighted later on in the results sections, all test RMSE and MAE are computed independently for each hydraulic variable. Accordingly, during evaluation, the MAE of the different hydraulic variables are never computed together and, as such, there is no need to perform a weighted average among them.

We corrected lines 283-288 as:

"We evaluated the performance using the multi-step-ahead RMSE (eq. (14)) over the whole simulation. However, for testing, we calculated the RMSE for each hydraulic variable o independently as:
$$RMSE_o = \frac{1}{H}\sum_{\tau=1}^{H}\|\hat{u}_o^\tau - u_o^\tau\|_2 \qquad\qquad (15)$$
Analogously, we evaluated the mean average error (MAE) for each hydraulic variable o over the whole simulation as:
$$MAE_o = \frac{1}{H}\sum_{\tau=1}^{H}\|\hat{u}_o^\tau - u_o^\tau\|_1 \qquad\qquad (16)"$$